 eLife

# Anti-nociceptive action of peripheral mu-opioid receptors by G-beta-gamma protein-mediated inhibition of TRPM3 channels

Sandeep Dembla[1†], Marc Behrendt[1†], Florian Mohr[1], Christian Goecke[1], Julia Sondermann[2], Franziska M Schneider[1], Marlene Schmidt[1], Julia Stab[3‡], Raissa Enzeroth[1], Michael G Leitner[1], Paulina Nuñez-Badinez[4], Jochen Schwenk[5], Bernd Nürnberg[6], Alejandro Cohen[7], Stephan E Philipp[3], Wolfgang Greffrath[4], Moritz Bünemann[8], Dominik Oliver[1], Eleonora Zakharian[9], Manuela Schmidt[2], Johannes Oberwinkler[1]*

[1]Institut für Physiologie und Pathophysiologie, Philipps-Universität Marburg, Marburg, Germany; [2]Max-Planck-Institut für Experimentelle Medizin, Göttingen, Germany; [3]Experimentelle und Klinische Pharmakologie und Toxikologie, Universität des Saarlandes, Homburg, Germany; [4]Department of Neurophysiology, Center of Biomedicine and Medical Technology Mannheim, Medical Faculty Mannheim Heidelberg University, Mannheim, Germany; [5]Institute of Physiology, Faculty of Medicine, University of Freiburg, Freiburg, Germany; [6]Abteilung für Pharmakologie und Experimentelle Therapie, Institut für Experimentelle und Klinische Pharmakologie und Toxikologie, Universität Tübingen, Tübingen, Germany; [7]Proteomics and Mass Spectrometry Core Facility, Life Sciences Research Institute, Dalhousie University, Halifax, Nova Scotia, Canada; [8]Institut für Pharmakologie und Klinische Pharmazie, Philipps-Universität Marburg, Marburg, Germany; [9]Department of Cancer Biology and Pharmacology, University of Illinois College of Medicine, Peoria, United States

*For correspondence:
johannes.oberwinkler@uni-marburg.de

†These authors contributed equally to this work

Present address: ‡Fraunhofer Institute for Biomedical Engineering, Sulzbach, Germany

Competing interests: The authors declare that no competing interests exist.

**Abstract** Opioids, agonists of μ-opioid receptors (μORs), are the strongest pain killers clinically available. Their action includes a strong central component, which also causes important adverse effects. However, μORs are also found on the peripheral endings of nociceptors and their activation there produces meaningful analgesia. The cellular mechanisms downstream of peripheral μORs are not well understood. Here, we show in neurons of murine dorsal root ganglia that pro-nociceptive TRPM3 channels, present in the peripheral parts of nociceptors, are strongly inhibited by μOR activation, much more than other TRP channels in the same compartment, like TRPV1 and TRPA1. Inhibition of TRPM3 channels occurs via a short signaling cascade involving Gβγ proteins, which form a complex with TRPM3. Accordingly, activation of peripheral μORs in vivo strongly attenuates TRPM3-dependent pain. Our data establish TRPM3 inhibition as important consequence of peripheral μOR activation indicating that pharmacologically antagonizing TRPM3 may be a useful analgesic strategy.

DOI: https://doi.org/10.7554/eLife.26280.001

**eLife digest** There are very few treatments available for people suffering from strong or long-lasting pain. Currently, substances called opioids – which include the well-known drug morphine – are the strongest painkillers. However, these drugs also cause harmful side effects, which makes them less useful.

Like all drugs, opioids mediate their effects by interacting with molecules in the body. In the case of opioids, these interacting molecules belong to a group of receptor proteins called G-protein coupled receptors (or GPCRs for short). These opioid receptors are widely distributed in the nerve cells and brain regions that detect and transmit pain signals. It was poorly understood how activation of opioid receptors reduces the activity of pain-sensing nerve cells, however several lines of evidence had suggested that a protein called TRPM3 might be involved.

TRPM3 is a channel protein that allows sodium and calcium ions to enter into nerve cells by forming pores in cell membranes, and mice that lack this protein are less sensitive to certain kinds of pain. Dembla, Behrendt et al. now show that activating opioid receptors on nerve cells from mice, with morphine and a similar substance, rapidly reduces the flow of calcium ions through TRPM3 channels. Further experiments confirmed that activating opioid receptors in a mouse's paw also reduced the pain caused when TRPM3 proteins are activated.

GPCRs interact with a group of small proteins called G-proteins that, when activated by the receptor, split into two subunits. Based on studies with human kidney cells, Dembla, Behrendt et al. found the so-called G-beta-gamma subunit then carries the signal from the opioid receptor to TRPM3. Two independent studies by Quallo et al. and Badheka, Yudin et al. also report similar findings.

These new findings show that drugs already used in the treatment of pain can indirectly alter how TRPM3 works in a dramatic way. These results might help scientists to find drugs that work in a more direct way to dial down the activity of TRPM3 and to combat pain with fewer side effects. Though first it will be important to confirm these new findings in human nerve cells.
DOI: https://doi.org/10.7554/eLife.26280.002

## Introduction

Throughout the peripheral and central parts of the nociceptive system, μ-opioid receptors (μORs) are widely expressed and strongly control neuronal excitation (*Stein, 2016*). Agonists of μORs are the most potent analgesic drugs clinically available (*Pasternak and Pan, 2013*) and are therefore often prescribed for the treatment of severe pain. These opioid substances are especially effective against acute pain states, such as post-operative pain, but they are also used, more controversially, for the treatment of longer lasting or chronic pain (*Rowbotham et al., 2003*; *Chou et al., 2015*). Much of the controversy around opioids arises because these substances cause important unwanted effects, such as addiction, tolerance (*Volkow and McLellan, 2016*), opioid-induced hyperalgesia (*Roeckel et al., 2016*) and, when overdosed, respiratory depression (*Pattinson, 2008*). Because of this unfavorable profile of unwanted effects, clinically used opioids are often implicated in fatal over-dosing due to drug addiction or dosing accidents (*Compton et al., 2016*; *Ray et al., 2016*). While many actions of opioids are triggered by activation of μORs in the central nervous system, opioid receptors are also located on the peripheral nerve endings of nociceptor neurons (*Stein et al., 1990a*, *1990b*; *Stein, 2013*). Physiologically, in the skin, where many peripheral nociceptor nerve endings reside, opioid receptors are targeted by endogenous opioid substances, such as β-endor-phin, released in the periphery from immune cells (*Stein et al., 1990b*) or skin keratinocytes (*Ibrahim et al., 2005*; *Fell et al., 2014*).

Activation of peripheral opioid receptors can provide clinically meaningful analgesia (*Far-ley, 2011*; *Stein and Machelska, 2011*). On the contrary, inhibiting peripheral μORs by antagonist application increases pain (*Jagla et al., 2014*). Targeting peripheral μORs thus has been proposed as a strategy to provide analgesia with reduced adverse effects and an improved safety profile (*Stein et al., 2003*). An alternative strategy, in which not the μORs themselves but downstream effectors of μOR signaling pathways are targeted, may also prove to be beneficial. However, such

strategies have received less attention, partly because the downstream targets of peripheral µOR signaling are not well documented.

At central synapses, several intracellular mechanisms leading to reduced neuronal excitation during µOR activation have been worked out in considerable detail. Activation of µORs causes inhibition of L-type voltage-gated $Ca^{2+}$ channels (*Bourinet et al., 1996*; *Heinke et al., 2011*) and activation of several types of $K^+$ channels (*Law et al., 2000*; *Marker et al., 2005*). Importantly, also at several central synapses, µORs have been shown to interact with TRP channels, notably TRPV1 (recently reviewed by *Bao et al., 2015*). The interaction of TRPV1 and µOR occurs at multiple anatomical locations in the central nervous system and its physiological consequences and mechanisms are highly diverse (*Maione et al., 2009*; *Stein, 2016*). Proposed mechanisms range from influencing the cAMP-PKA pathway and the participation of several other kinases (ERK, MAPK, JNK) to β-arrestin2 and the PI3-kinase pathway (*Law et al., 2000*; *Rowan et al., 2014*; *Bao et al., 2015*).

In peripheral nerves and their endings, several, partially different, intracellular signaling events have been proposed to account for the effectiveness of peripherally restricted opioids, including inhibition of voltage-gated $Na^+$ channels (*Gold and Levine, 1996*), inhibition of HCN channels (*Ingram and Williams, 1994*), activation of several classes of $K^+$ channels (*Cunha et al., 2010*; *Nockemann et al., 2013*; *Baillie et al., 2015*), and, importantly, again the inhibition of TRPV1 channels (*Vetter et al., 2006*; *Endres-Becker et al., 2007*; *Spahn et al., 2013*) However, despite the plethora of proposed targets, the magnitude and the interplay of these individual effects has not been determined. Furthermore, many of these ion channels are not inhibited by µOR activation per se, but rather their upregulation or sensitization by the cAMP/PKA pathway is blocked by µOR signaling. It is therefore unclear, how much each of these targets contributes to the overall effect of opioids on peripheral nerve endings, and how this contribution may vary during shifts from resting to inflamed states (and back). Further, it is unclear, and perhaps even unlikely, that all important downstream targets of µORs have already been identified.

Recently, TRPM3 channels (*Oberwinkler and Philipp, 2014*) were described in primary nociceptive neurons (*Nealen et al., 2003*; *Lechner et al., 2009*; *Vriens et al., 2011*; *Straub et al., 2013a, 2013b*; *Held et al., 2015*; *Usoskin et al., 2015*) and it was shown that these divalent-permeable cation channels (*Grimm et al., 2003*; *Oberwinkler et al., 2005*; *Wagner et al., 2010*) play an important role in noxious heat sensation since these channels are intrinsically thermosensitive (*Vriens et al., 2011*). Activation of TRPM3 leads to release of pro-inflammatory CGRP from peripheral nociceptor nerve terminals (*Held et al., 2015*), while TRPM3-deficient animals show severe defects in the development of inflammatory hyperalgesia (*Vriens et al., 2011*). In this context, it is interesting that inhibitors of TRPM3 channels have been identified that exhibit strong anti-nociceptive properties (*Straub et al., 2013b*; *Chen et al., 2014*; *Suzuki et al., 2016*; *Krügel et al., 2017*). Hence, TRPM3 channels in peripheral nociceptors have pro-nociceptive and pro-inflammatory properties, making them an interesting target to study the mechanism of peripheral nociception and inflammation. Here, we demonstrate that TRPM3 channels in primary nociceptive neurons are rapidly and strongly inhibited by agonists of µORs such as morphine, through an intracellular signaling cascade relaying µORs to TRPM3 channels via Gβγ proteins. We show that local activation of peripheral µORs causes strong analgesia of TRPM3-dependent pain. Our results indicate that it may be worthwhile to further investigate TRPM3 channels as potential targets for the treatment of pain with reduced adverse effects in the central nervous system.

## Results

### Activation of µORs inhibits TRPM3 channels in DRG neurons

TRPM3 channels possess pro-nociceptive properties in DRG neurons (*Vriens et al., 2011*; *Straub et al., 2013a, 2013b*; *Chen et al., 2014*; *Held et al., 2015*). We therefore investigated clinically used analgesic drugs for their effects on TRPM3 channels. To address this question, we first activated TRPM3 channels in isolated mouse DRG neurons by applying the known TRPM3 agonist pregnenolone sulfate (PS) (*Wagner et al., 2008*; *Vriens et al., 2011*). Under our recording conditions, the ensuing $Ca^{2+}$ signals were almost completely dependent on the presence of TRPM3 proteins, as demonstrated in control experiments with neurons prepared from TRPM3-deficient (TRPM3-knockout) mice (*Figure 1—figure supplement 1*). Testing whether morphine, a traditional

agonist of opioid receptors, or its synthetic, μOR-specific analog DAMGO (*Handa et al., 1981*), influences TRPM3 channel activity, we found that both μOR agonists strongly reduced the $Ca^{2+}$ signals induced by PS (*Figure 1a–c*). The reduction of these $Ca^{2+}$ signals was rapid and readily reversible within a time frame of 1–2 min (*Figure 1d*). We ascertained that DAMGO acted via activating opioid receptors by co-applying the opioid receptor antagonist naloxone, which unspecifically blocks several opioid receptor subtypes (*Figure 1e–g*). Naloxone alone did not affect TRPM3-dependent $Ca^{2+}$ signals in DRG neurons (*Figure 1e,g*), but, when co-applied with DAMGO, it prevented the action of DAMGO completely (*Figure 1f,g*). In addition, we tested two further, structurally different, μOR agonists, herkinorin and loperamide. Both substances strongly inhibited PS-induced $Ca^{2+}$ signals in isolated mouse DRG neurons (*Figure 1—figure supplement 2*). These data indicate that activation of μORs, independent of the chemical nature of the agonist, leads to inhibition of TRPM3 channels in DRG neurons.

We successfully reconstituted the signaling pathway from μORs to TRPM3 channels in a heterologous overexpression system by co-expressing μORs and TRPM3 in HEK cells (*Figure 1h*). Importantly, omitting μOR cDNA from the transfection abolished the effect of morphine (*Figure 1h*) and DAMGO (as seen, for example, in Figure 6f), providing strong evidence that these agonists act through μORs and do not interfere with TRPM3 channel activity directly. Like many members of the TRP superfamily, TRPM3 channels are activated by several different agonists (*Oberwinkler and Philipp, 2014*). Taking advantage of the overexpression system, which avoids the complication of other nifedipine- or heat-sensitive conductances (*Fajardo et al., 2008*; *Julius, 2013*), we demonstrated that TRPM3 channels activated by the application of nifedipine (*Wagner et al., 2008*; *Drews et al., 2014*) or by heat applied in the absence of chemical agonists (*Vriens et al., 2011*) were also inhibited by μOR activation (*Figure 1i,j*).

Whole-cell patch-clamp electrophysiological recordings from HEK cells overexpressing TRPM3 and μORs demonstrated that currents through TRPM3 channels were inhibited by activating μORs rapidly in less than 5 s (*Figure 1k–m*, *Figure 1—figure supplement 3*). After measuring a dose-response curve by varying the concentration of DAMGO, we found that the $IC_{50}$ values at +80 mV and −80 mV did not differ, providing an indication that inhibition of TRPM3 channels by μOR activation was voltage-independent (*Figure 1—figure supplement 4*). Similarly, performing patch-clamp experiments on isolated DRG neurons, we observed that the currents evoked by TRPM3 agonists were rapidly (less than 5 s) inhibited by μOR activation (*Figure 2a–c*). Upon washout of DAMGO, the inhibition was at least partially reversible. Activation of opioid receptors causes many cellular responses in DRG neurons (*Law et al., 2000*), including the activation of $K^+$ channels (*Yoshimura and North, 1983*; *Ocaña et al., 2004*). Importantly, stimulation of $K^+$ channels might reduce $Ca^{2+}$ signals in DRG neurons through hyperpolarization-induced closure of voltage-gated $Ca^{2+}$ channels. Because $K^+$ channels cannot directly be studied in $Ca^{2+}$ imaging experiments, we continued to use the whole-cell patch-clamp technique on isolated DRG neurons. Aiming for small-diameter nociceptors, we selected cells that responded either to TRPV1- or to TRPM3-activating (or to both types of) agonists. In these cells, we could not detect outward currents upon application of the μOR agonist DAMGO (*Figure 2d–f*). This indicates that the reduction of PS-induced currents by μOR activation is not due to the activation of outward currents. Consequently, the DAMGO-induced reduction of PS-evoked $Ca^{2+}$ signals observed in $Ca^{2+}$ imaging experiments (e.g. in *Figure 1a–g*) is unlikely to be caused by $K^+$ channel-dependent hyperpolarization. Together, these data argue that TRPM3 channels in DRG neurons are inhibited by activating μORs.

## μOR-mediated inhibition of TRPM3 channels occurs mainly in TRPV1-expressing DRG neurons

Our patch-clamp data of PS-sensitive DRG neurons indicated that the inhibition of TRPM3-dependent currents in DRG neurons was variable (*Figure 2c*). We therefore measured a large number of DRG neurons in $Ca^{2+}$ imaging experiments (*Figure 2g–i*) and found equally that not in all PS-sensitive neurons activation of μORs caused a reduction of $Ca^{2+}$ signals. When we categorized these neurons, the DAMGO-reactive subset amounted to 91% (562 cells) in a total of 614 PS-sensitive cells analyzed (*Figure 2i*). We tested whether TRPM3 inhibition by μOR activation correlated with the presence of functional TRPV1 channels in these cells, the rational of this being that in adult mice, TRPV1 channels are restricted to peptidergic nociceptors (*Cavanaugh et al., 2011*), in which μORs are also preferentially expressed (*Vetter et al., 2006*; *Scherrer et al., 2009*). In agreement with this,

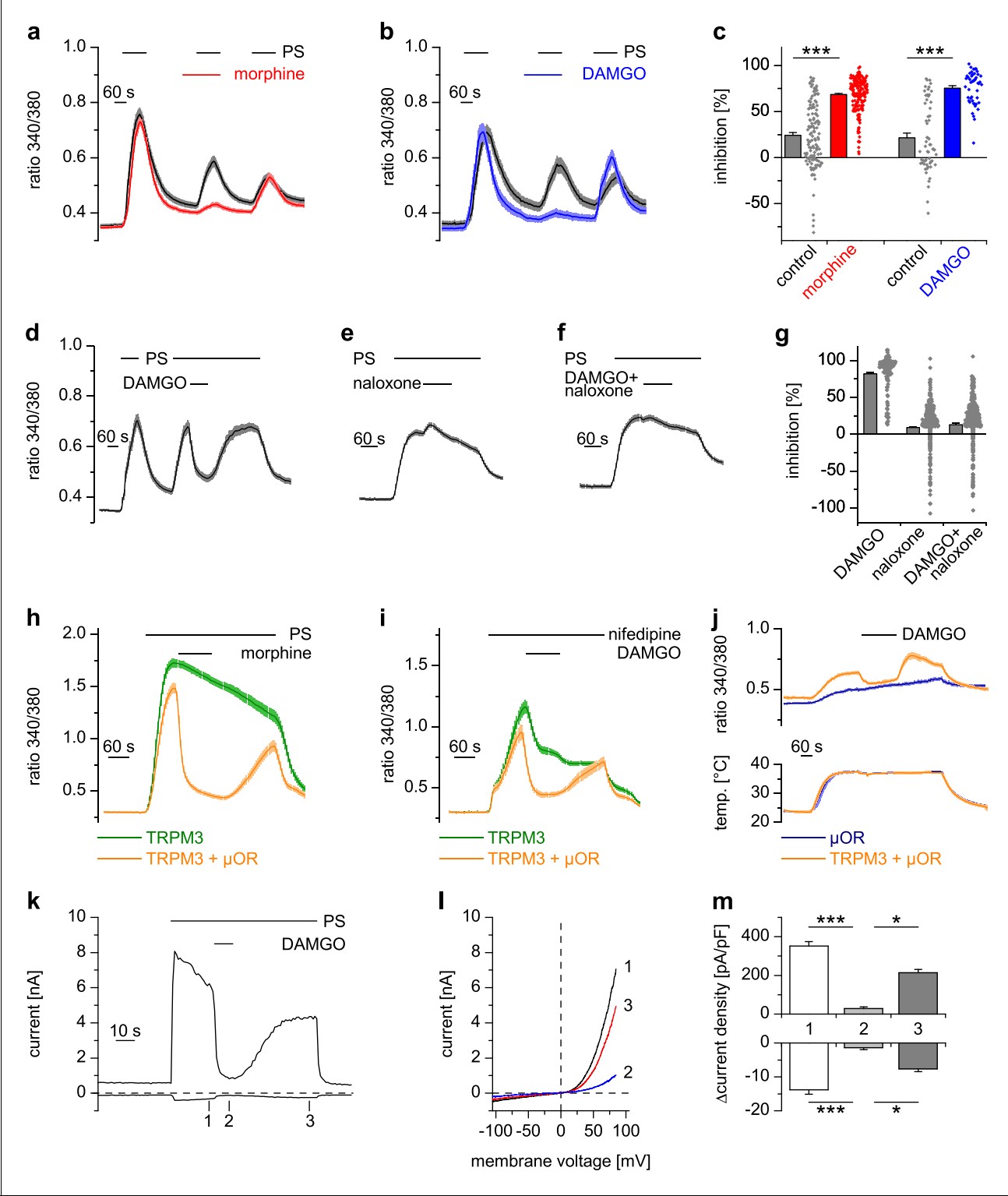

**Figure 1.** Activated μORs inhibit TRPM3-induced Ca$^{2+}$ signals in somatosensory neurons and in an overexpression system. (a) DRG neurons were stimulated three times with PS (25 μM) and morphine (1 μM, red trace, n = 188 neurons, five recordings) or vehicle as control (black trace, n = 130 neurons, four recordings). (b) Instead of morphine, DAMGO (0.3 μM, blue trace, 47 neurons, three recordings) was applied (control, black trace, 59 neurons, three recordings). Images of cells isolated from DRGs as used in these experiments are presented in *Figure 1—figure supplement 5*, example traces of individual cells in *Figure 1—figure supplement 6a*. (c) Quantification of the inhibition measured in (a) and (b); each dot-like symbol next to the bars represents the inhibition measured and calculated for a single, individual cell. Control experiments showing that the recorded Ca$^{2+}$

*Figure 1 continued on next page*

*Figure 1 continued*

signals are TRPM3-dependent are presented in *Figure 1—figure supplement 1*. Similar data with two chemically distinct µOR agonists are shown in *Figure 1—figure supplement 2*. (d) Inhibition by 0.3 µM DAMGO and recovery in DRG neurons stimulated with 25 µM PS, 0.3 µM DAMGO as indicated (n = 111 neurons, seven recordings). (e) The opioid receptor antagonist naloxone (5 µM) did not have an effect on its own (n = 412 neurons, 14 recordings). (f) Naloxone blocked the inhibitory effect of DAMGO on TRPM3 (n = 364 neurons, 15 recordings). Example traces of individual cells from these experiments are shown in *Figure 1—figure supplement 6b*. (g) Statistical analysis of the recordings shown in panels (d–f). The inhibition caused by DAMGO or naloxone was evaluated. The symbols on the right side of the bars represent inhibition measured in individual cells. (h) In HEK cells overexpressing TRPM3 and µORs, morphine (1 µM) inhibited $Ca^{2+}$ signals evoked by 25 µM PS (orange trace, n = 115 cells, four recordings), but not in control cells expressing only TRPM3 (green trace, n = 97 cells, same four recordings). (i) DAMGO (0.3 µM) inhibits TRPM3-dependent $Ca^{2+}$ signals induced by 50 µM nifedipine (orange trace: n = 58 cells, green trace: n = 75 cells, both from three recordings). (j) DAMGO (0.3 µM) inhibits TRPM3-dependent $Ca^{2+}$ signals induced by heat (upper panel). Orange trace (n = 59 cells from four recordings) is the average of cells overexpressing TRPM3 and µORs. The blue trace (n = 89 cells, three recordings) represents control measurements from HEK cells expressing only µORs; there, TRPM3-independent heat-evoked $Ca^{2+}$ signals are not inhibited by µOR activation. The lower panel shows the time course of the applied temperature. (k) Exemplary patch-clamp recording of a HEK cell overexpressing TRPM3 (activated by 25 µM PS) and µORs (activated by 0.3 µM DAMGO). Traces were obtained at +80 and −80 mV from voltage ramps. (l) I/V-curves measured at time points indicated in (k). (m) Statistical analysis of the baseline-subtracted current densities (n = 15 cells). A graphical representation of the recorded values for each cell is given in *Figure 1—figure supplement 3*. A dose response curve for the DAMGO-induced inhibition of TRPM3 currents is shown in *Figure 1—figure supplement 4*.

DOI: https://doi.org/10.7554/eLife.26280.003

The following figure supplements are available for figure 1:

**Figure supplement 1.** Pregnenolone sulfate (PS)-induced $Ca^{2+}$ signals are dependent on TRPM3.
DOI: https://doi.org/10.7554/eLife.26280.004
**Figure supplement 2.** The chemically different opioid receptor agonists herkinorin and loperamide also cause inhibition of TRPM3 channels.
DOI: https://doi.org/10.7554/eLife.26280.005
**Figure supplement 3.** Data of each individual cell analyzed for *Figure 1m*.
DOI: https://doi.org/10.7554/eLife.26280.006
**Figure supplement 4.** Dose-response relationship for the inhibition measured in a similar way as the recordings in *Figure 1k–m*.
DOI: https://doi.org/10.7554/eLife.26280.007
**Figure supplement 5.** Images of isolated DRG cells plated on a coverslip and loaded with Fura2.
DOI: https://doi.org/10.7554/eLife.26280.008
**Figure supplement 6.** Exemplary responses of single cells from the experiments shown in *Figure 1a–c* (a) or *Figure 1d–g* (b).
DOI: https://doi.org/10.7554/eLife.26280.009

we found that most neurons (465 of 562 cells, 83%), which showed a reduction of TRPM3-dependent $Ca^{2+}$ signals due to application of DAMGO, also showed capsaicin-induced $Ca^{2+}$ signals, demonstrating the presence of functional TRPV1 channels. Conversely, only a minority of the cells without apparent effect of DAMGO on TRPM3-dependent $Ca^{2+}$ signals (14 of 52 DAMGO-insensitive cells, i.e. 27% of the DAMGO-insensitive cells, corresponding to 2% of all PS-sensitive cells) showed a response to capsaicin (*Figure 2i*). These data are corroborated by the size distribution of both, DAMGO-sensitive and DAMGO-insensitive, subgroups of PS-sensitive cells (*Figure 2—figure supplement 1*). PS-sensitive cells of both subgroups showed the typical distribution of small-diameter C- and Aδ-type neurons (*Vriens et al., 2011*; *Tan and McNaughton, 2016*), with only very few larger cells (diameter >31 µm: 26 of 614 cells, equivalent to 4.2%). On the other hand, the group of neurons that responded neither to PS nor to capsaicin appeared to contain a somewhat higher number of larger cells (diameter >31 µm: 146 of 1673 cells, equivalent to 8.9%; *Figure 2—figure supplement 1c,d*). Together, these data show that activation of µORs is capable of rapidly and strongly reducing currents through TRPM3 channels in a subset of DRG neurons, corresponding largely, but not exclusively, to putative peptidergic nociceptors.

## TRPM3 channels are inhibited by receptors coupled to $G\alpha_{i/o}$-containing G proteins

DRG neurons express a host of different G-protein-coupled receptors (GPCRs) (*Gold and Gebhart, 2010*; *Veldhuis et al., 2015*), many of which have been implied in regulating the excitability of these neurons. µORs couple preferentially to members of the $G\alpha_i$ subfamily, i.e. $G_{i/o}$ proteins (*Law et al., 2000*). In general, GPCRs that couple to $G_{i/o}$ proteins have been implied in dampening the activity of peripheral nociceptors. We therefore tested other GPCRs that couple to $G\alpha_i$-containing G proteins and found that all of them inhibited TRPM3-dependent $Ca^{2+}$ signals in DRG neurons, although

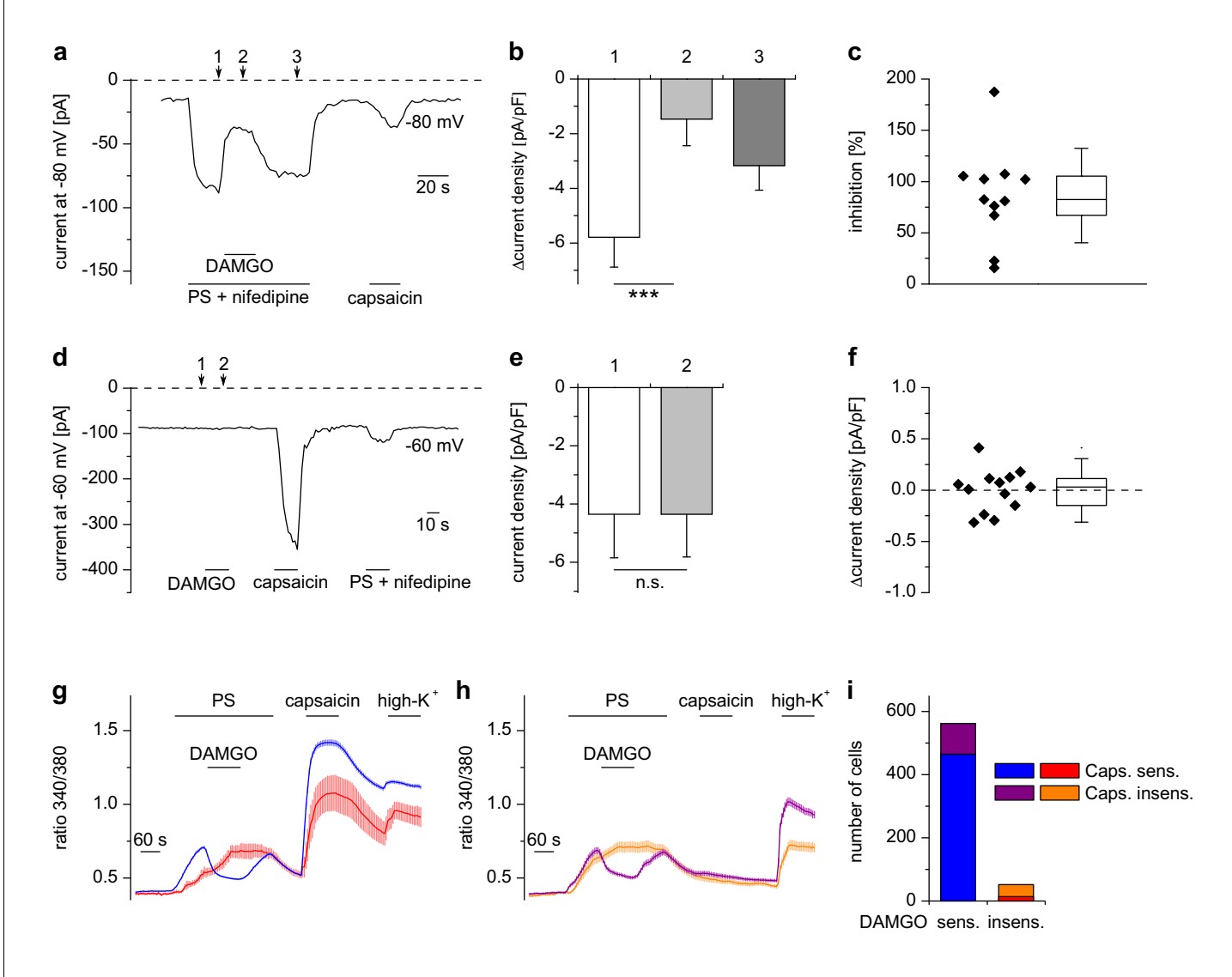

**Figure 2.** Inhibition of TRPM3 by μORs in DRG neurons is variable. (a) Example of an electrophysiological whole-cell patch-clamp recording from an isolated DRG neuron responding to TRPM3 agonists (PS and nifedipine at 50 μM each), DAMGO (0.3 μM) and capsaicin (0.1 μM). The recording was performed in monovalent-free solution in order to reduce currents through endogenous voltage-gated channels. (b) Statistical analysis of 11 cells responding with an inward current to the combined application of PS and nifedipine. Baseline-subtracted current densities were evaluated at similar time points as indicated by arrows and numbers in (a). (c) To illustrate the variability, the inhibition observed in individual cells is shown. (d–f) Application of DAMGO alone (in standard extracellular solution containing K$^+$) did not evoke significant currents in DRG neurons responsive to TRPM3 or TRPV1 agonists. (d) Exemplary recording of a neuron at −60 mV holding potential sensitive to 1 μM capsaicin and TRPM3 agonists (PS and nifedipine at 50 μM each), (e) statistical analysis of 13 different DRG neurons, (f) DAMGO-induced increase in current density in individual cells, evaluated as difference in current density for each cell between time points 1 and 2 marked with arrows in (d). In (c) and (f), the box indicates mean and 25 and 75 percentiles, the whiskers represent standard deviations. (g–i) Similarly, in Ca$^{2+}$ imaging experiments, Ca$^{2+}$ signals evoked by 25 μM PS in DRG neurons were variably sensitive to inhibition by DAMGO (3 μM). PS-sensitive DRG neurons (17% of total neurons, i.e. 614 out of 3576 neurons, obtained from 5 mice in 35 recordings) were classified as sensitive (g) or insensitive (h) to 2 μM capsaicin, and each subgroup was further divided into DAMGO-sensitive and DAMGO-insensitive cell populations and averaged separately. (i) Overview of the number of cells in each subgroup (which also corresponds to the number of neurons averaged for the traces in (g) and (h)). Overall, 91.5% (562 of 614) of the PS-sensitive neurons were inhibited by DAMGO. The size distribution of the cells analyzed in (g–i) is shown in *Figure 2—figure supplement 1*. A graphical representation of the response of the individual cells analyzed in this experiment is given in *Figure 2—figure supplement 2*.

DOI: https://doi.org/10.7554/eLife.26280.010

The following figure supplements are available for figure 2:

**Figure supplement 1.** DAMGO-sensitive and DAMGO-insensitive TRPM3-expressing neurons have similar size distributions.

*Figure 2 continued on next page*

*Figure 2 continued*

DOI: https://doi.org/10.7554/eLife.26280.011

**Figure supplement 2.** Responses of individual cells from the experiments shown in *Figure 2g–i*.

DOI: https://doi.org/10.7554/eLife.26280.012

the number of neurons responding to the specific agonists was variable (*Figure 3*). Specifically, we found that TRPM3 was inhibited by activating GABA$_B$ receptors, CB$_2$ cannabinoid receptors, δ-opioid receptors, adrenoreceptors and somatostatin receptors (*Figure 3*). When we activated metabotropic glutamate type five receptors (mGluR5), which preferentially couple to Gα$_q$ proteins (*Bhave et al., 2001*) we observed the weakest inhibition of all agonists, which was only detectable in a relatively small number of neurons. In addition, the weak inhibition caused by DHPG did not recover upon washout (*Figure 3f*). These data show that the activity of TRPM3 channels can be inhibited by a large variety of Gα$_i$-coupled GPCRs.

## Activation of μORs does not strongly inhibit other TRP channels expressed in DRG neurons

Apart from TRPM3, other TRP channels are involved in temperature sensing and inflammatory processes in peripheral somatosensory neurons (*Julius, 2013*). We tested therefore whether other TRP channels prominently expressed in DRG neurons were inhibited by activation of μORs, by using the same experimental conditions as on TRPM3 channels. Ca$^{2+}$ signals induced by the TRPV1 agonist capsaicin showed the same reduction in activity whether DAMGO or the solvent control (DMSO) was applied to the DRG neurons (*Figure 4a,b*). Furthermore, when we used a different protocol and co-applied DAMGO together with longer applied capsaicin, there was a much weaker inhibition compared to the inhibition typically observed with TRPM3 (*Figure 4—figure supplement 1*). Notably, we did not observe any difference when we differentiated between PS-sensitive and PS-insensitive neurons (*Figure 4—figure supplement 1a*, green vs red trace), although 91% of the PS-sensitive neurons are expected to possess functional μORs (see *Figure 2i*). Additionally, the capsaicin-induced Ca$^{2+}$ signals did not recover after DAMGO washout (*Figure 4—figure supplement 1a*), again in marked contrast to the behavior of TRPM3 channels. We then investigated the sensitivity of TRPA1 channels on μOR activation in DRG neurons. Because TRPA1 channels are difficult to stimulate several times, we used the second type of protocol. Similarly to our results with TRPV1, we observed that AITC-induced Ca$^{2+}$ signals were not subject to inhibition by DAMGO, regardless of whether the neurons were PS-sensitive or not (*Figure 4c,d*). When we overexpressed TRPV1 channels together with μORs in HEK cells, we found no evidence for any inhibitory action of μORs on TRPV1 channels in the time frame investigated (2 min; *Figure 4e*). Similarly, we investigated the coupling of μORs to TRPA1 and TRPM8 channels in overexpression systems, again without finding any indication of inhibitory action of μORs on these channels (*Figure 4f,g*). In summary, from the group of TRP channels implicated in noxious thermoreception and inflammatory hyperalgesia, only TRPM3 channels displayed a substantial, rapid and reversible inhibition during μOR activation.

## The influence of μORs on TRPM3 channel activity is dependent on G$_{i/o}$ proteins, but does not involve changes in cAMP concentration

Usually μORs couple to G$_{i/o}$ proteins, but also other signaling pathways have been described for these receptors (*Law et al., 2000*). We tested whether G$_{i/o}$ proteins were involved in the functional coupling of μORs to TRPM3 channels by incubating cultured DRG neurons for 16–24 hr with pertussis toxin (PTX), which selectively disrupts signaling via G$_{i/o}$ proteins (*Mangmool and Kurose, 2011*). We found that this treatment strongly reduced the action of DAMGO on PS-evoked activity in DRG neurons (*Figure 5a,b*). Moreover, in HEK cells overexpressing μORs and TRPM3 proteins, PTX treatment almost completely abrogated the DAMGO-induced inhibition of TRPM3 channels (*Figure 5c, d*). These data strongly implicate the classical signaling pathway comprising G proteins containing Gα$_{i/o}$ subunits in the inhibition of TRPM3.

Activated Gα$_i$ proteins reduce the activity of adenylyl cyclases and may thereby lower the concentration of cytosolic cAMP. We tested whether this process is required for inhibiting TRPM3 channels. We found that application of IBMX (to inhibit cAMP-degrading phosphodiesterases) and forskolin

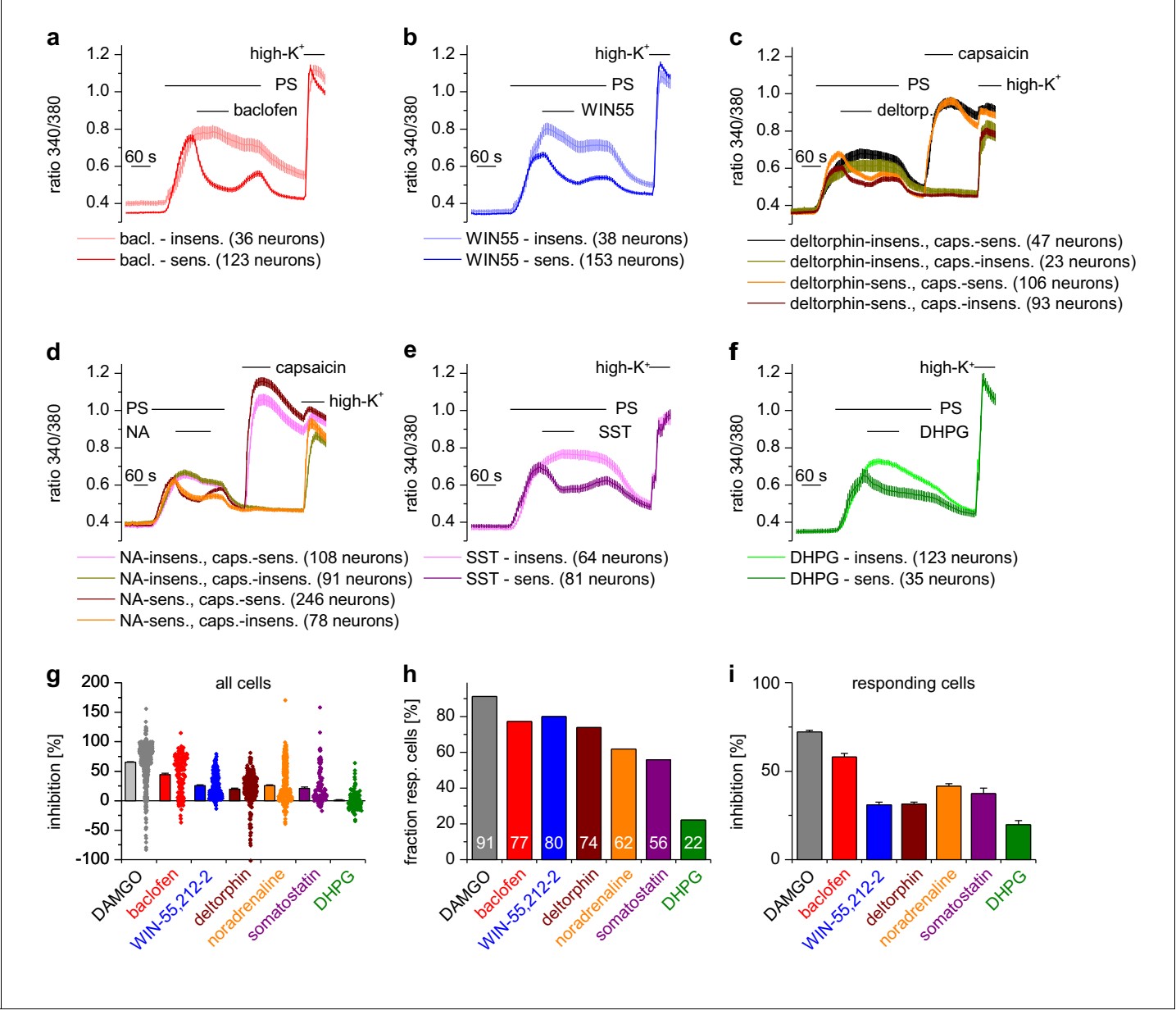

**Figure 3.** A variety of different Gα$_i$-coupled receptor types inhibits TRPM3 channels in isolated DRG neurons. Neurons sensitive to pregnenolone sulfate (PS, 25 µM) were selected post-hoc and categorized as sensitive or insensitive with respect to the GPCR agonist used. In (c) and (d), neurons were additionally subdivided in groups according to capsaicin (2 µM) sensitivity, similar to the analysis in *Figure 2g–i*. Traces represent averages within these categories, numbers in brackets give the number of neurons used for the analysis. (a) 100 µM baclofen (bac) was used to stimulate GABA$_B$ receptors (four recordings, cells from one mouse). (b) 1 µM WIN-55,212–2 (WIN55) was used to stimulate cannabinoid receptors (four recordings, cells from one mouse). (c) 1 µM deltorphin (deltorp.) was used to stimulate δ-opioid receptors (18 recordings, cells from three mice). (d) 2 µM noradrenaline (NA) was used to stimulate adrenoreceptors (17 recordings, cells from two mice). (e) 0.5 µM somatostatin (SST) was used to activate somatostatin receptors (eight recordings, cells from two mice). (f) Gα$_q$-coupled mGluR5 receptors were stimulated with 20 µM dihydroxyphenylglycine (DHPG, four recordings, cells from one mouse). Traces of individual cells from the subgroups shown in panels (a) – (f) are shown in *Figure 3—figure supplement 1*. (g) Percentage of inhibition of the PS-evoked Ca$^{2+}$ signal of all PS-sensitive neurons. The dot-like symbols on the right side of the columns represent the inhibition measured for each individual cell. (h) Summary of the experiments showing the percentage of PS-sensitive neurons responding with a reduction of PS-induced Ca$^{2+}$ signals (larger than the predefined threshold of 7.5%, see Materials and methods section) to the stimulation with the indicated agonist. The numerical value of the percentage of GPCR-agonist-sensitive cells is indicated on the columns. (i) Same as panel (g), but calculating the average inhibition of only those cells that show an inhibition larger than the predefined threshold value. The data for the µOR agonist DAMGO in panels (g–i) are taken from *Figure 2g–i*. Combined, these data indicate that the inhibition of TRPM3-dependent signals is especially pronounced when stimulating µORs (with DAMGO) or GABA$_B$ receptors (with baclofen).

*Figure 3 continued on next page*

*Figure 3 continued*

DOI: https://doi.org/10.7554/eLife.26280.013

The following figure supplement is available for figure 3:

**Figure supplement 1.** Ca²⁺ responses of 10 individual, randomly chosen, neuronal DRG cells for each group of the experiments shown in *Figure 3*.

DOI: https://doi.org/10.7554/eLife.26280.014

(to stimulate cAMP-producing adenylyl cyclases) did not influence TRPM3 channel activity in DRG neurons nor their inhibition by μOR activation (*Figure 5e,f*). We observed the same outcome in TRPM3- and μOR-overexpressing HEK cells after combined application of forskolin and IBMX (*Figure 5g,h*). Changing concentrations of cAMP might influence the activity of protein kinase A (PKA). However, when we dialyzed TRPM3- and μOR-expressing HEK cells with the unhydrolyzable ATP analog AMP-PNP through the pipette in patch-clamp experiments, effectively blocking the action of all kinases, including PKAs, we found no evidence that this treatment affected the inhibitory action of μOR activation on TRPM3 channels (*Figure 5i,k*). On the other hand, when we removed Mg²⁺ ions from the intracellular solution, the inhibiting action of DAMGO on TRPM3 channel activity was lost after 5 min of internally dialyzing the cells via the patch pipette (*Figure 5j,k* and *Figure 5— figure supplement 1a*), showing that under our recording conditions the exchange between pipette solution and the cytosol was sufficiently rapid. These data also support our earlier conclusion that μOR-mediated TRPM3 inhibition relies on G-protein signaling as G proteins need Mg²⁺ ions to bind GTP (*Gilman, 1987*). We made sure that the series resistance during the patch-clamp recordings was not significantly different between the experimental groups (*Figure 5—figure supplement 1b*). Therefore, these experiments suggest that kinases are not involved in the signaling pathway of μORs to TRPM3 channels. This conclusion was further corroborated by experiments using kinase inhibitors (H89, a relatively non-specific kinase inhibitor (*Lochner and Moolman, 2006*); KT5720, inhibitor of PKA; BIM, inhibitor of PKC), none of which abolished the inhibition of TRPM3 channels by μOR activation (*Figure 6*). However, some of these pharmacological kinase inhibitors, in particular H89 and KT5720, had unexpected and potentially unspecific effects on the measured Ca²⁺ signals and on TRPM3 activation (*Figure 6* and *Figure 6—figure supplement 1*). We did not further investigate these effects. Taken together, these data (*Figures 5* and *6*, *Figure 5—figure supplement 1*) strongly indicate that TRPM3 inhibition after μOR activation is a G-protein-coupled process, but does not involve the second messenger cAMP or downstream kinases.

## The inhibitory action of μORs on TRPM3 channels is dependent on G protein βγ subunits

Heterotrimeric G proteins consist of α and dimeric βγ subunits and both of them can act as intracellular messengers (*Wettschureck and Offermanns, 2005*). We overexpressed Gα or Gβγ subunits together with TRPM3 channels in HEK cells and assessed their activity by applying PS. Overexpression of each of the three Gα$_i$ proteins, either in wild-type form (*Figure 7a*), or as constitutively active QtoL mutant (*Graziano and Gilman, 1989*) of Gα$_{i1}$ (*Figure 7b*) had no inhibitory effect on the activity of TRPM3 channels. We also tested YFP-tagged Gα$_i$ subunits (*Figure 7—figure supplement 1a*), because these allowed to easily assess the expression of the proteins by fluorescence microscopy. These tagged Gα$_i$ subunits also failed to inhibit TRPM3 channels. Additionally, we assessed the expression of Gα$_{i3}$ and the YFP-tagged Gα$_i$ subunits by Western blotting and found them to be well expressed in HEK cells (*Figure 8—figure supplement 3a*). Since the proteins were detected in these blots at the location of their expected size and since we did not detect obvious signs of proteolysis, we concluded that Gα$_i$ subunits do not inhibit TRPM3 activity. Equally to the Gα$_i$ subunits, overexpression of Gα$_{o1}$ (*Figure 7c*) or Gα$_{o2}$ (*Figure 7d*) did not affect PS-induced TRPM3 channel activity. Finally, also overexpression of Gα$_q$ proteins, which recently have been shown to influence TRPM8 channels (*Zhang et al., 2012*), did not reduce the TRPM3-evoked Ca²⁺ signals (*Figure 7—figure supplement 1b*).

However, when we overexpressed Gβ$_1$ and Gγ$_2$ proteins together with TRPM3 channels, TRPM3-dependent Ca²⁺ signals were strongly reduced (*Figure 7e*). Importantly, the reduction of TRPM3 activity by Gβγ subunits was attenuated by the additional overexpression of (myristoylated) myr-βARKct (*Koch et al., 1994*; *Rishal et al., 2005*), a Gβγ-binding peptide, indicating

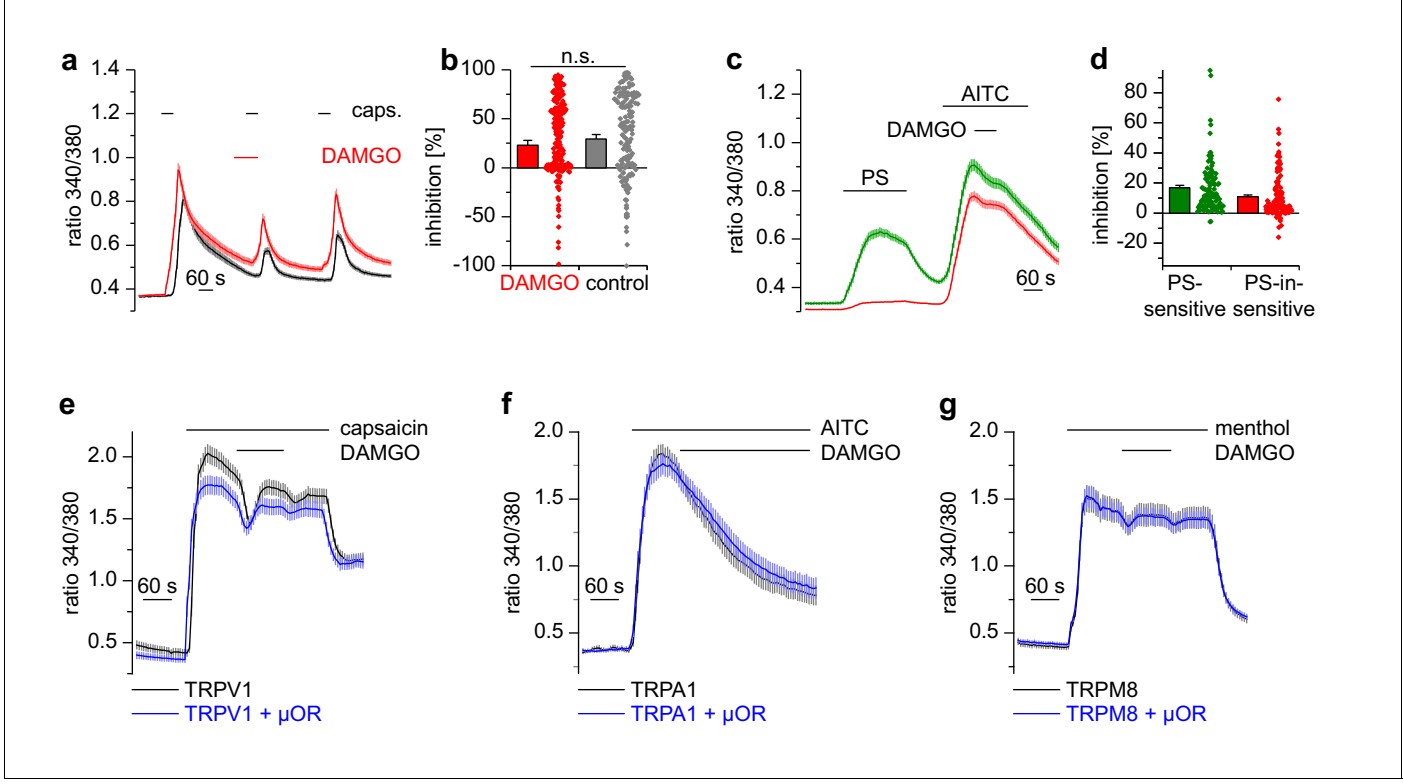

**Figure 4.** Activation of µORs does not rapidly inhibit the somatosensory temperature-sensitive TRP channels TRPV1, TRPA1 and TRPM8. (**a**) The capsaicin-sensitive subset of DRG neurons showed no reduction of the capsaicin-evoked $Ca^{2+}$ signals (0.1 µM capsaicin, red trace: n = 166 neurons from 10 recordings, cells from two mice) after application of 0.3 µM DAMGO that was stronger than the reduction observed during the application of the solvent (control trace: 0.015% DMSO, black trace: n = 186 neurons from nine recordings, cells from the same two mice). (**b**) Statistical summary of the inhibition evoked by DAMGO or DMSO in the experiment in (**a**). The dot-like symbols on the right side of the bars indicate inhibition values measured in individual cells. Please note, that for both measurement conditions ('DAMGO' and 'control'), two cells showing strongly increased responses during the application of DAMGO or DMSO (corresponding to inhibition values smaller than −100%) are not shown in the scatter plots. These data, however, were included in the calculation of mean and SEM as shown in the bars. (**c**) With a different application protocol, AITC-sensitive DRG neurons were selected and tested for inhibition of the AITC-evoked $Ca^{2+}$ signals by DAMGO (3 µM). Averaged traces show that the AITC-evoked $Ca^{2+}$ signals were not strongly inhibited by DAMGO, regardless of whether they were PS-sensitive (green trace: n = 107 neurons) or not (red trace: n = 132 neurons, recordings of both traces come from the same eight recordings, all cells were from one mouse). (**d**) Statistical summary of the experiments in (**c**). The dot-like symbols in (**b**) and (**d**) on the right of the columns represent inhibition values of single cells. Data obtained for TRPV1 channels with the same stimulation protocol as in (**c**) are shown in *Figure 4—figure supplement 1*. Examples of recordings from individual DRG neurons are shown in *Figure 4—figure supplement 2*. (**e**) HEK cells overexpressing TRPV1 and µORs were exposed to 0.1 µM capsaicin and 3 µM DAMGO (blue trace, n = 81 cells, three recordings). Control cells were only transfected with TRPV1 (black trace, n = 94 cells, three recordings). (**f**) In similar experiments CHO cells expressing TRPA1 and µORs were exposed to 50 µM AITC and 3 µM DAMGO (blue trace, n = 53 cells, two recordings). In control experiments, CHO cells expressing only TRPA1 were treated identically (black trace, n = 65 cells, two recordings). (**g**) In further experiments, HEK cells expressing TRPM8 and µORs (blue trace, n = 103 cells, two recordings) or only TRPM8 (black trace, n = 64 cells, two recordings) were exposed to 200 µM menthol and 3 µM DAMGO. In (**e–g**), application of DAMGO did not lead to a reduction of $Ca^{2+}$ signals evoked by the specific agonists of TRPV1, TRPA1 and TRPM8, indicating that these channels were not inhibited by activated µORs.

DOI: https://doi.org/10.7554/eLife.26280.015

The following figure supplements are available for figure 4:

**Figure supplement 1.** Activation of µORs in TRPV1-expressing neuronal DRG cells causes only little inhibition of TRPV1 channels.
DOI: https://doi.org/10.7554/eLife.26280.016

**Figure supplement 2.** Example traces from individual DRG neurons of experiments shown in *Figure 4*.
DOI: https://doi.org/10.7554/eLife.26280.017

that the inhibition of TRPM3 channels was caused specifically by the overexpressed Gβγ subunits in an unbound state (*Figure 7f,g*). A suppression of TRPM3 activity by overexpressed Gβγ subunits could also be demonstrated in whole-cell patch-clamp recordings of transfected HEK cells (*Figure 7h,i*).

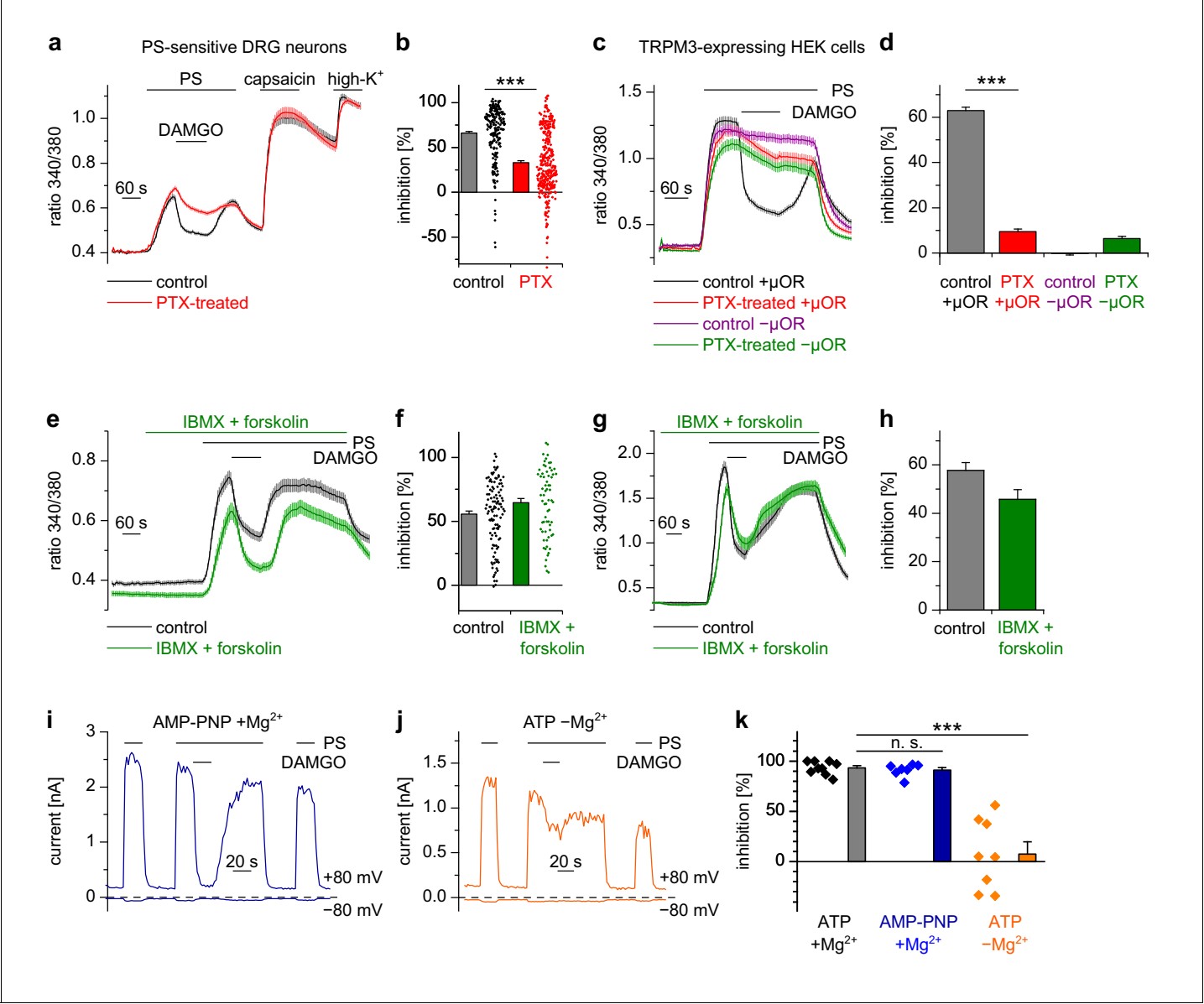

**Figure 5.** Inhibition of TRPM3 by activated μORs is dependent on Gα$_{i/o}$ proteins, but not on cAMP or kinases. (a) Isolated DRG neurons were treated with PTX (red trace, n = 319 neurons, 14 recordings) or not treated (black trace, n = 211 neurons, seven recordings) and tested as indicated with PS (25 μM), DAMGO (3 μM) and capsaicin (2 μM). Only PS-sensitive neurons were analyzed (all cells were from three mice). (b) Statistical summary: PTX treatment significantly reduced the inhibitory effect of DAMGO. (c) In similar experiments, treatment with PTX (red trace; n = 159 cells, three recordings) abolished the inhibition of TRPM3 channels in HEK cells overexpressing TRPM3 and μORs (control black trace, n = 178 cells, three recordings). The violet trace represents additional control cells untreated with PTX that expressed TRPM3, but not μORs (n = 163 cells, same three recordings as for the black trace), the green trace represents cells treated with PTX but not transfected with μORs (n = 142 cells, same three recordings as for the red trace). (d) Statistical summary of the experiments shown in (c). (e, f) Application of forskolin (10 μM) together with IBMX (200 μM) did not prevent the action of DAMGO (0.3 μM) in isolated DRG neurons (green trace: n = 67 neurons, treated with IBMX +forskolin, three recordings; black trace: n = 114 neurons, untreated, from three recordings). In this panel, all cells are from one mouse. Example traces of individual cells of the experiments shown in (a) and (e) are shown in *Figure 5—figure supplement 2*. (g, h) The same result was obtained in HEK cells overexpressing TRPM3 and μORs (green trace: n = 76 treated cells; black trace: n = 91 untreated cells, five recordings for each condition). (i) In whole-cell patch-clamp recordings of HEK cells overexpressing TRPM3 and μORs, the inhibition of PS-activated TRPM3-dependent currents (25 μM PS) by 3 μM DAMGO was almost complete, even when intracellular ATP was replaced by the non-hydrolyzable analog AMP-PNP. (j) However, removal of intracellular Mg$^{2+}$ abolished the inhibition by DAMGO. two exemplary recordings are shown in (i, j), an exemplary control measurement is shown in *Figure 5—figure supplement 1a*. In these recordings, the break-in to the whole-cell configuration occurred 200 s before the beginning of the traces shown (i.e. 300 s before the application of DAMGO). (k) Quantitative analysis of the inhibition of 9 cells under control (with ATP and with Mg$^{2+}$) conditions, 7 cells with AMP-PNP (and Mg$^{2+}$) and 8 cells without Mg$^{2+}$ (but with ATP) at membrane potentials of +80 mV. *Figure 5—figure supplement 1b* demonstrates that

*Figure 5 continued on next page*

Figure 5 continued

the series resistance of the recordings was not statistically different between the experimental groups analyzed here. In (b), (f) and (k), each individual symbol represents the value obtained from a single cell.

DOI: https://doi.org/10.7554/eLife.26280.018

The following figure supplements are available for figure 5:

**Figure supplement 1.** Control experiments for measurements in *Figure 5i–k*.

DOI: https://doi.org/10.7554/eLife.26280.019

**Figure supplement 2.** Responses of individual neurons from the experiments shown in *Figure 5*.

DOI: https://doi.org/10.7554/eLife.26280.020

We next tried to increase the concentration of free Gβγ subunits with a different approach that does not necessitate transfections and overexpression of exogenous proteins. The peptide mSIRK, when applied extracellularly, enters the cells and induces the dissociation of heterotrimeric G proteins without inducing GDP/GTP exchange in the Gα subunits (*Goubaeva et al., 2003*). Using this approach, we observed a significant reduction in TRPM3 activity in DRG neurons as well as in TRPM3 overexpressing HEK cells (*Figure 7—figure supplement 3*). The inactive analog mSIRK-L9A did not cause this effect. However, mSIRK also induced an increased $Ca^{2+}$ concentration in the cells at baseline, which was especially prominent in DRG neurons. It is therefore unclear if the reduction in TRPM3 activity was caused by the increased concentration of free Gβγ subunits, or if it was caused more indirectly by the increased free $Ca^{2+}$ concentration. When we tried to manipulate the concentration of free Gβγ subunits in the opposite direction by overexpression of Gβγ-binding peptides, myr-βARKct as before or myr-phosducin (*Schulz, 2001*; *Rishal et al., 2005*), we found that the inhibitory signaling from μORs to TRPM3 channels was severely reduced (*Figure 7—figure supplement 2*). In aggregate, these data strongly suggest a model in which TRPM3 channels are inhibited by free Gβγ dimers, but not by $Gα_{i/o}$ subunits. Interestingly, in the experiments where non-active $Gα_{i/o}$ subunits were overexpressed (*Figure 7a–d*, *Figure 7—figure supplement 1a*), we consistently observed an increase of the resting $Ca^{2+}$ concentration. Possibly, this increase is due to an increased basal activity of TRPM3 channels caused by binding and thereby scavenging of free Gβγ subunits by the overexpressed Gα subunits. To test this speculation, further experimental work will be necessary.

## TRPM3 and Gβ proteins are parts of the same protein complex

To examine whether TRPM3 proteins and Gβγ subunits form a protein complex, we immunoprecipitated TRPM3 proteins equipped with a C-terminal YFP tag with anti-GFP antibodies (which also recognize YFP). In control experiments, we determined that fused tags (YFP or myc) on TRPM3 proteins do not interfere with the functional coupling of μORs to TRPM3 channels, irrespective of their N- or C-terminal location (*Figure 8—figure supplement 1*). After separating the precipitated proteins, we observed a single band in western blots at the molecular weight expected for myc-TRPM3-YFP using antibodies against GFP, indicating that our procedure indeed precipitated full-length TRPM3 proteins (*Figure 8a*). When we probed blots with antibodies against Gβ subunits, we found evidence for Gβ co-precipitating with TRPM3 from HEK-TRPM3 cells. Using the same protocol on HEK cells without TRPM3 did not lead to precipitation of Gβ proteins (*Figure 8b,c*, *Figure 8—figure supplement 2*). We further tested the ability of the anti-Gβ antibodies to recognize Gβ proteins by overexpressing Gβ proteins with a FLAG tag and therefore increased molecular weight and successfully detected the predicted bands in Western blots (*Figure 8—figure supplement 3b*). Finally, in order to ascertain the identity of Gβ proteins interacting with TRPM3 without the potentially confounding issue of antibody specificity, we identified in mass spectrometry experiments peptide fragments belonging unequivocally to $Gβ_1$ proteins. These peptides originated from trypsin-digested SDS gels containing proteins co-immunoprecipitating with TRPM3 from HEK-TRPM3 cells (*Figure 9*).

Although $Gα_{i3}$ subunits were abundantly present in the cell lysates, we could not detect them on western blots of proteins co-precipitating with TRPM3 (*Figure 8d*; see also *Figure 8—figure supplement 3a* for tests of the antibodies against $Gα_{i3}$). Because TRPM8 proteins have recently been shown to interact directly with $Gα_q$ proteins (*Zhang et al., 2012*), we also tested for, but failed to detect $Gα_q$ proteins co-precipitating with TRPM3 (*Figure 8e*). These data show that TRPM3 and Gβ

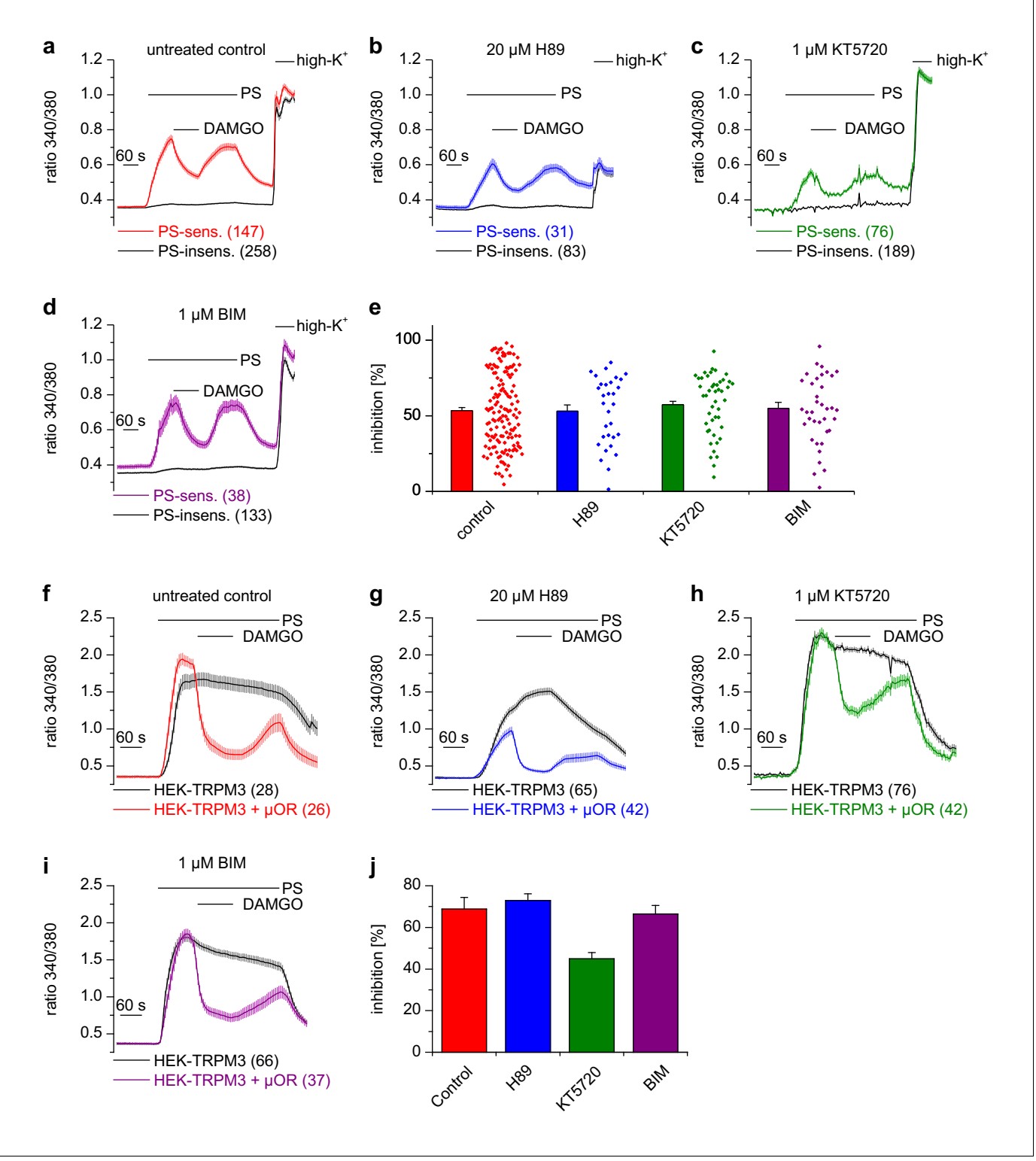

**Figure 6.** Inhibition of TRPM3 channels is not blocked by kinase inhibitors. (a–e) Isolated DRG neurons were treated with kinase inhibitors for 30 min before the response to 25 µM pregnenolone sulfate (PS) and its inhibition by 0.3 µM DAMGO was assayed. Colored traces represent PS-sensitive, black traces PS-insensitive neurons, all treated as indicated (control cells were exposed to the equivalent amount of vehicle, 0.1% DMSO). In panels (a–d), the number of neurons averaged is indicated in brackets, they were isolated from two mice and measured in 3 (panels b–d) or six recordings (panel a). (e) Quantitative summary of the inhibition induced by DAMGO shows that treatment with kinase inhibitors did not impede the inhibitory action of µORs. The dot-like symbols on the right side of the bars indicate inhibition measured in individual cells. Example traces of individual cells are shown in

*Figure 6 continued on next page*

Figure 6 continued

*Figure 6—figure supplement 1*. (f–i) Similar experiments on HEK cells overexpressing TRPM3 and μORs (colored traces) or only TRPM3 (black traces). The number of cells averaged is indicated in brackets. They were measured in three (panel **f**) or four (panels **g–i**) recordings. (j) Quantitative summary showing that the treatment with the indicated kinase inhibitors did not prevent the inhibitory action of activated μORs. Unspecific effects on TRPM3 and other $Ca^{2+}$ channels were observed, especially with 20 μM H89 (**b, g**), which, however, does not change the main conclusion that kinases are not involved in the inhibitory actions of μORs on TRPM3 channels.
DOI: https://doi.org/10.7554/eLife.26280.021
The following figure supplement is available for figure 6:

**Figure supplement 1.** Responses of individual cells (10 in each panel) under one of the four treatment conditions from *Figure 6a–e*.
DOI: https://doi.org/10.7554/eLife.26280.022

proteins are present in the same protein complexes. This provides a mechanistic explanation for the inhibitory action of μORs exerted on TRPM3 channels.

## TRPM3-dependent, but not TRPV1-dependent, pain is alleviated by activation of peripheral μORs

We injected either PS, which has been shown to evoke nocifensive behavior and thus pain in a strictly TRPM3-dependent manner (*Vriens et al., 2011*; *Straub et al., 2013a*), or capsaicin to evoke TRPV1-dependent pain into the hind paws of mice and observed the duration of the ensuing nocifensive behavior. Co-injection of DAMGO to activate peripheral μORs (*Stein and Lang, 2009*) strongly reduced the nocifensive behavior evoked by PS injection, but did not affect the TRPV1-dependent pain after capsaicin injection (*Figure 10a,b*). Injecting higher concentrations of capsaicin into the hind paw produced considerably longer responses, showing that the lower capsaicin dose used in *Figure 10b* did not saturate the pain-evoked behavior under our conditions (*Figure 10c*). Again, co-injecting DAMGO was ineffective in reducing the duration of the capsaicin-induced nocifensive behavior at these higher capsaicin doses (*Figure 10c*). These data show that locally, in the peripheral skin of living mice, TRPM3 channels, much more than TRPV1 channels, are under rapid inhibitory control of μORs.

## Discussion

Both, endogenously released and therapeutically applied opioids, acting through peripherally located μORs, effectively mediate pain relief (*Farley, 2011*; *Stein, 2016*). We show here that TRPM3 channels, expressed in primary nociceptive neurons, are a major target of an intracellular signaling cascade initiated by activated μORs. TRPM3 channels are inhibited by activated μORs strongly, rapidly and, reversibly (*Figures 1* and *2a–c*).

### Regulation of TRPM3 by GPCRs

These results add TRPM3 channels to the category of TRP channels expressed in primary somatosensory neurons that are under tight regulatory control of GPCRs and their associated signaling cascades (*Veldhuis et al., 2015*). The only other intracellular regulatory mechanism affecting TRPM3 channels uncovered so far is the hydrolysis of $PIP_2$ (*Badheka et al., 2015*; *Tóth et al., 2015*; *Uchida et al., 2016*), which typically is brought about by activating $G\alpha_q$-coupled receptors. In the panel of receptors that we tested here, we also included a $G\alpha_q$-coupled receptor (mGluR5), for which we found that it induces only weak and inconsistent inhibition of TRPM3 that did not recover (*Figure 3f–i*, *Figure 3—figure supplement 1f*). There are several reasons possible why mGluR5 seems not to couple strongly to TRPM3, but it should be noted that DRG neurons express many more $G\alpha_q$-coupled receptors, for which these results cannot necessarily be extrapolated. It is still largely unexplored and therefore entirely unclear whether $PIP_2$ metabolism induced by $G\alpha_q$-coupled receptor activation is physiologically relevant for the regulation of TRPM3 channels in primary nociceptor neurons.

### Regulation of TRP channels by $G\beta\gamma$ proteins

Mechanistically, the combined data from pharmacological experiments, overexpression studies and biochemical interaction analyses presented here strongly suggest a model in which TRPM3 channel

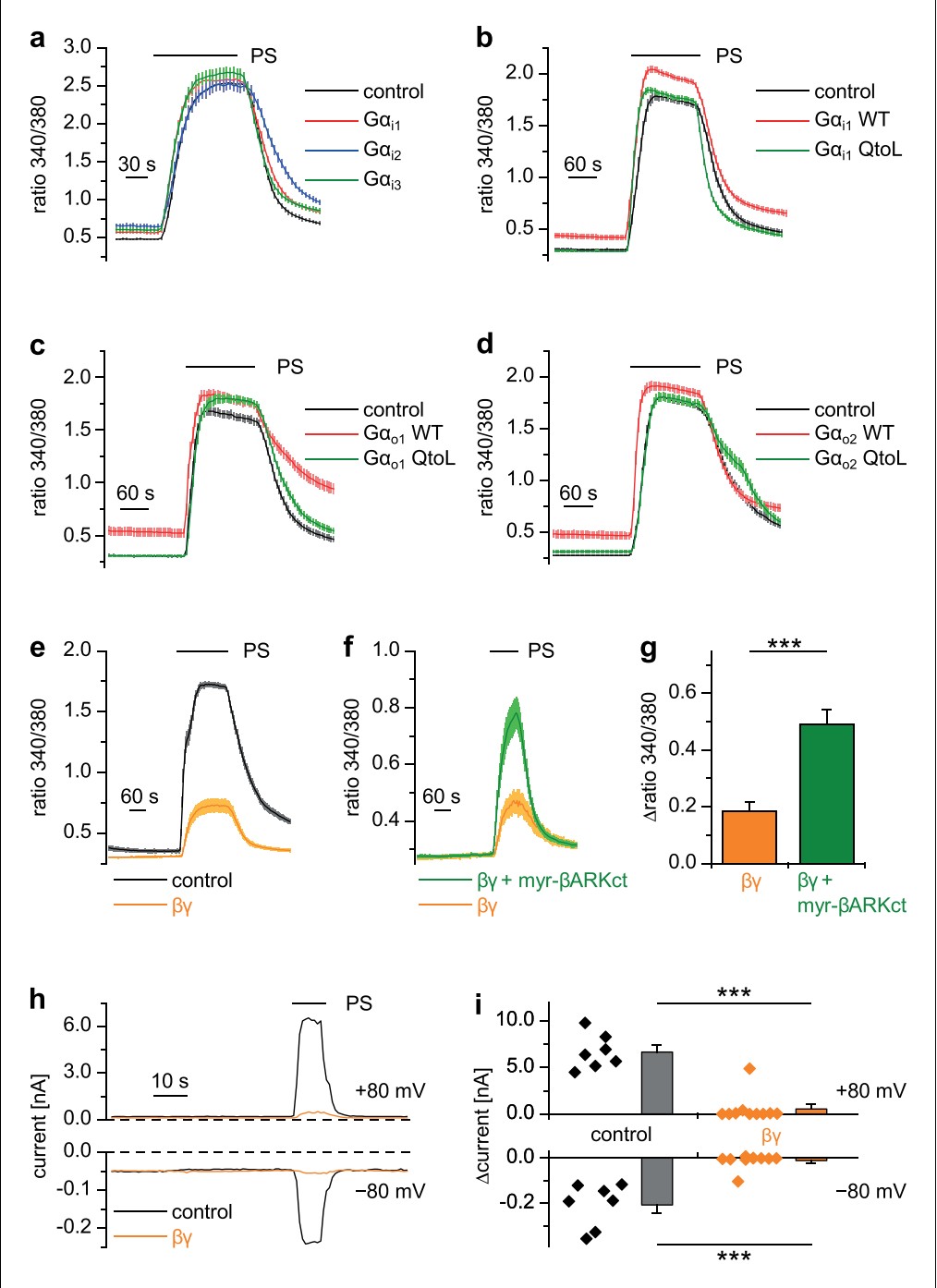

**Figure 7.** Gβγ dimers, but not the α$_i$ or α$_o$ subunits of G proteins inhibit TRPM3 channels. (**a–d**) TRPM3-expressing HEK cells were transfected with various G protein subunits (and Ds-Red for marking successfully transfected cells): (**a**) Gα$_{i1}$ (red trace: n = 178 cells, four recordings), Gα$_{i2}$ (blue trace: n = 123 cells, three recordings) or Gα$_{i3}$ (green trace: n = 147 cells, four recordings; black control trace: n = 124 cells, four recordings), (**b**) Gα$_{i1}$ as wild-type protein (red trace: n = 189 cells, four recordings) or with the Q204L (QtoL) mutation that renders the Gα subunit constitutively active (green trace: n = 128 cells, three recordings; black control trace: n = 92 cells, two recordings), (**c**) with wild-type Gα$_{o1}$ (red trace: n = 75 cells, three recordings) or constitutive active (QtoL mutated) Gα$_{o1}$ (green trace: 87 cells, four recordings; black control trace: 77 cells, three recordings), or (**d**) with wild-type Gα$_{o2}$ (red trace: 101 cells, three recordings) or with QtoL mutated Gα$_{o2}$ (green trace: 75 cells, three recordings; control cells shown as black trace: 114 cells, three recordings). In these experiments, control cells were mock-transfected with a vector expressing only GFP. None of the overexpressed Gα proteins reduced the PS-induced TRPM3 activation.

*Figure 7 continued on next page*

*Figure 7 continued*

Experiments with overexpression of additional Gα subunits are presented in *Figure 7—figure supplement 1*. (e) In contrast, overexpressing Gβ$_1$ and Gγ$_2$ proteins in TRPM3-expressing HEK cells strongly reduced TRPM3 channel activity (orange trace: n = 68 cells, four recordings; control cells were mock-transfected, black trace: n = 72 cells, four recordings). (f) This inhibition was reduced by concomitantly overexpressing Gβγ-scavenging myristoylated βARKct (green trace: n = 70 cells, four recordings, compared to the orange trace: n = 41 cells, three recordings). (g) Statistical analysis of the experiment shown in (f). (h) Exemplary whole-cell patch-clamp recordings showing the reduction of PS-induced TRPM3 channel activity in cells transfected with Gβ$_1$ and Gγ$_2$. (i) Statistical analysis of 7 control cells and 10 cells transfected with Gβ$_1$ and Gγ$_2$, evaluated at two different holding potentials (+80 and −80 mV). The effect of Gβγ-scavengers on μOR-induced inhibition of TRPM3 was also investigated and the results are shown in *Figure 7—figure supplement 2*. The effect of application of mSIRK (which releases Gβγ without receptor activation) on TRPM3-dependent Ca$^{2+}$ signals is shown in *Figure 7—figure supplement 3*.
DOI: https://doi.org/10.7554/eLife.26280.023

The following figure supplements are available for figure 7:

**Figure supplement 1.** Overexpression of additional Gα proteins did not inhibit TRPM3 channels.
DOI: https://doi.org/10.7554/eLife.26280.024

**Figure supplement 2.** Expression of phosducin or βARKct reduced the inhibition of TRPM3 channels induced by the activation of μORs.
DOI: https://doi.org/10.7554/eLife.26280.025

**Figure supplement 3.** Interfering with Gβγ proteins through exogenous peptides influences TRPM3 activity.
DOI: https://doi.org/10.7554/eLife.26280.026

---

complexes are inhibited by directly binding to Gβγ dimers, but not to Gα subunits. The key evidence for this assertion is that (1) overexpression of Gβγ, but not Gα proteins inhibits TRPM3 channels (*Figure 7*), that (2) TRPM3 proteins are found in the same protein complexes as Gβ proteins (*Figures 8* and *9*) and that (3) alternative pathways involving cAMP and protein kinases could be ruled out (*Figure 5e–k* and *Figure 6*). Still other pathways that are entirely independent of heterotrimeric G proteins also cannot account for our findings, because sensitivity to PTX strongly implicates G proteins from the Gα$_i$ subfamily (*Figure 5a–d*). Binding to Gβγ is the mechanism with which μORs affect many other ion channels, like voltage-gated Ca$^{2+}$ and GIRK channels (*Bourinet et al., 1996*; *Law et al., 2000*; *Marker et al., 2005*; *Heinke et al., 2011*). In the family of TRP ion channels, however, direct regulation by Gβγ appears not to be common. One report linked TRPA1 channel activation to Gβγ dimers liberated after activation of the GPCR MrgprA3 (*Wilson et al., 2011*). We tested therefore, whether μOR activation has an effect on TRPA1 channels in DRG neurons, but observed neither an inhibition nor a pronounced activation of these channels. This finding could indicate that μORs and TRPA1 channels are expressed in largely separate subpopulations of DRG neurons. However, we also failed to observe an effect of μOR activation in a heterologous overexpression system where μOR activation had a clear effect on TRPM3 (*Figure 4f*), indicating that the regulation of TRPA1 channels by Gβγ proteins might not be direct and requires further investigation. The only other TRP channel that has been reported to be regulated by Gβγ dimers is TRPM1 (*Shen et al., 2012*), the channel protein with the highest homology to TRPM3. TRPM1, however, is not known to be expressed in somatosensory neurons. The assertion that TRPM1 is inhibited by Gβγ dimers has been controversial since earlier work reported that purified Gβγ dimers are ineffective, while Gα$_o$ proteins inhibit heterologously expressed TRPM1 (*Koike et al., 2010*). A recent publication apparently reconciles these findings by providing evidence that both G protein subunits bind to TRPM1 and inhibit the channel (*Xu et al., 2016*). Still, it is an open question, which of these G protein entities is more important in regulating TRPM1 channels under physiological conditions.

In addition to the proposed regulation of TRPM1 by Gα$_o$ proteins, TRPM8 has recently been shown to directly bind Gα$_q$ proteins and to be strongly inhibited by this event (*Zhang et al., 2012*). It was therefore important to test whether TRPM3 binds to and whether it is regulated by Gα subunits. Our data indicate that neither is the case (*Figures 7* and *8*), reinforcing our model that the main signaling pathway from μOR to TRPM3 channels is via Gβγ proteins.

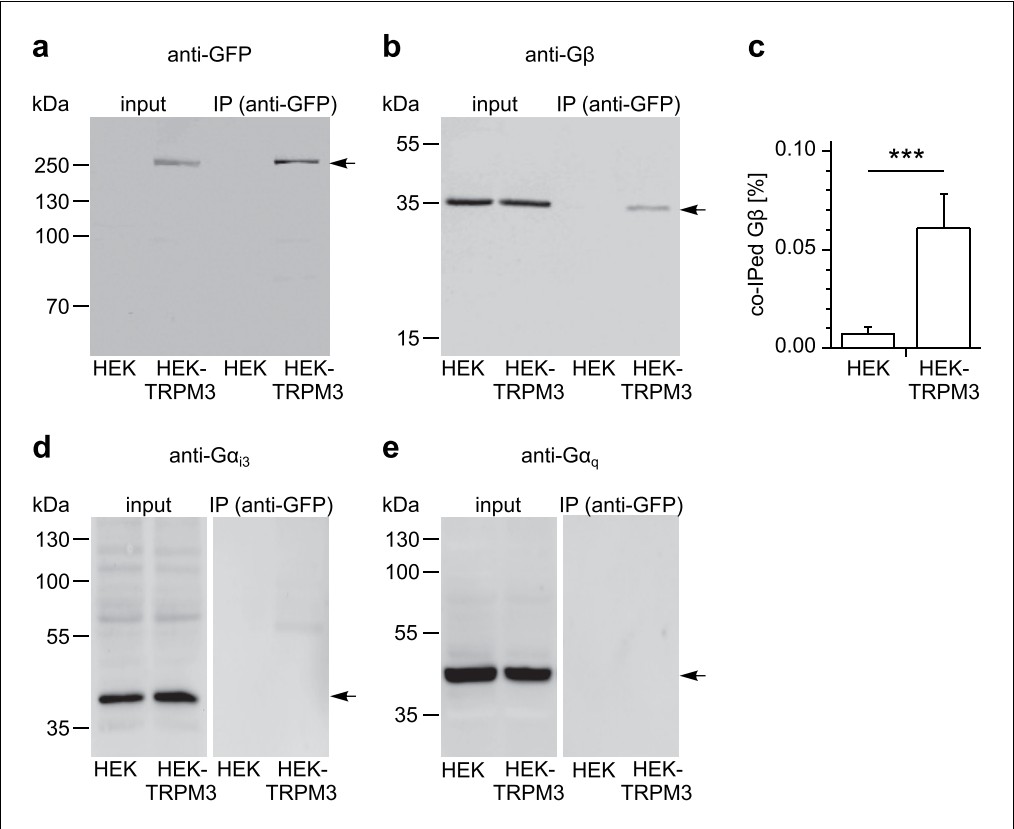

**Figure 8.** TRPM3 proteins form a complex with β subunits of G proteins. In co-immunoprecipitation experiments, myc-TRPM3-YFP was precipitated from transfected HEK cells with anti-GFP coated beads (in control experiments we first determined that these tags do not interfere with TRPM3 function, see *Figure 8—figure supplement 1*). Untransfected HEK cells were used as control. After separation by SDS-PAGE and western-blotting, proteins were detected with antibodies. Antibodies against (**a**) GFP (these antibodies also detect YFP, one representative blot out of four is shown), (**b**) Gβ (one of seven blots is shown), (**d**) Gα$_{i3}$ (one of four blots is shown) or (**e**) Gα$_q$ (one of two blots is shown) were used. Input represents a defined total lysate fraction. (**c**) After densitometric quantification of immunoprecipitation experiments similar to those shown in (**b**), the amount of co-immunoprecipitated Gβ subunits was normalized to the total amount of Gβ in the lysate and averaged (data were obtained from seven individual blots originating from seven independent experiments). Significantly more Gβ protein was precipitated from cells expressing myc-TRPM3-YFP than from untransfected control cells. The densitometric values for each individual blot are shown in *Figure 8—figure supplement 2*. Additional blots testing the specificity of antibodies against Gα$_{i3}$ and Gβ are presented in *Figure 8—figure supplement 3*.
DOI: https://doi.org/10.7554/eLife.26280.027

The following figure supplements are available for figure 8:

**Figure supplement 1.** N- or C-terminal tags on TRPM3 proteins do not compromise their functional properties.
DOI: https://doi.org/10.7554/eLife.26280.028
**Figure supplement 2.** Densitometric data from individual western blots used for the statistical analysis in *Figure 8c*.
DOI: https://doi.org/10.7554/eLife.26280.029
**Figure supplement 3.** Western blots used for testing of antibody specificity and degradation of G proteins.
DOI: https://doi.org/10.7554/eLife.26280.030

## Signaling pathways employed by peripheral μORs

The intracellular pathways by which peripheral μORs reduce the excitability of primary nociceptive neurons seem diverse. A PI3Kγ/NO pathway has been described for the stimulation of K$_{ATP}$ channels, where PI3Kγ is activated by Gβγ subunits (*Cunha et al., 2010*). Also, GIRK2 channels activated directly by Gβγ proteins have been implicated in peripheral opioid analgesia (*Nockemann et al., 2013*). These channels are absent from mouse DRG neurons, but may play a role in human anti-

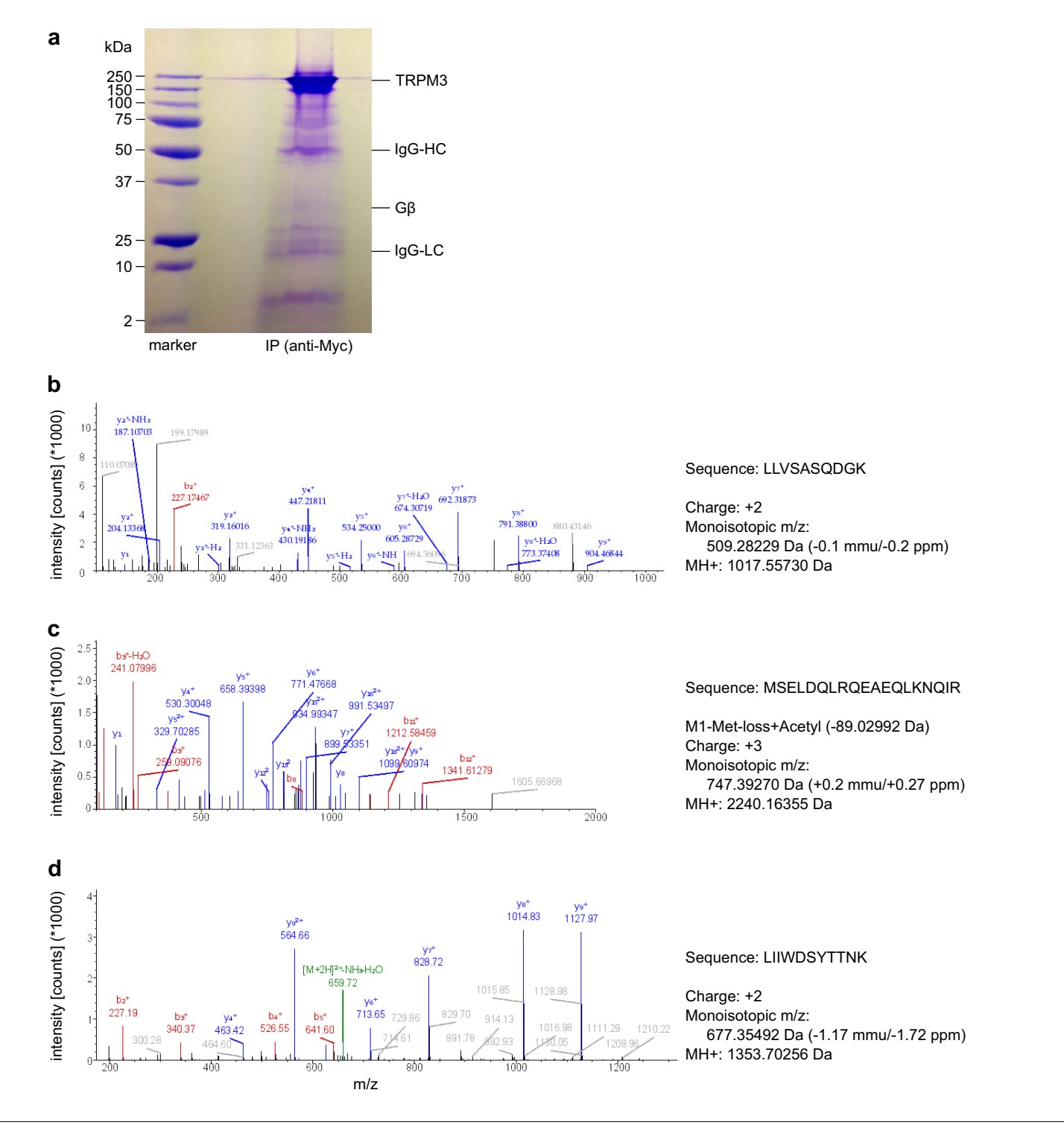

**Figure 9.** Mass spectrometry confirms the identity of Gβ₁ proteins as co-immunoprecipitating with TRPM3 proteins. (**a**) Coomassie blue stained SDS-polyacrylamide gel showing the proteins co-immunoprecipitated from HEK cells stably expressing myc-tagged TRPM3 using anti-myc antibodies coupled to magnetic beads. The bands corresponding to TRPM3, Gβ proteins, and the heavy and light chains of the antibodies are indicated. A representative gel of the two gels analyzed is shown. (**b–d**) Spectra from three different peptides originating from Gβ₁ proteins co-immunoprecipitated with TRPM3. In the two different co-immunoprecipitation experiments, the peptide shown in (**b**) was found twice, the peptide in (**c**) four times and the peptide in (**d**) once.

DOI: https://doi.org/10.7554/eLife.26280.031

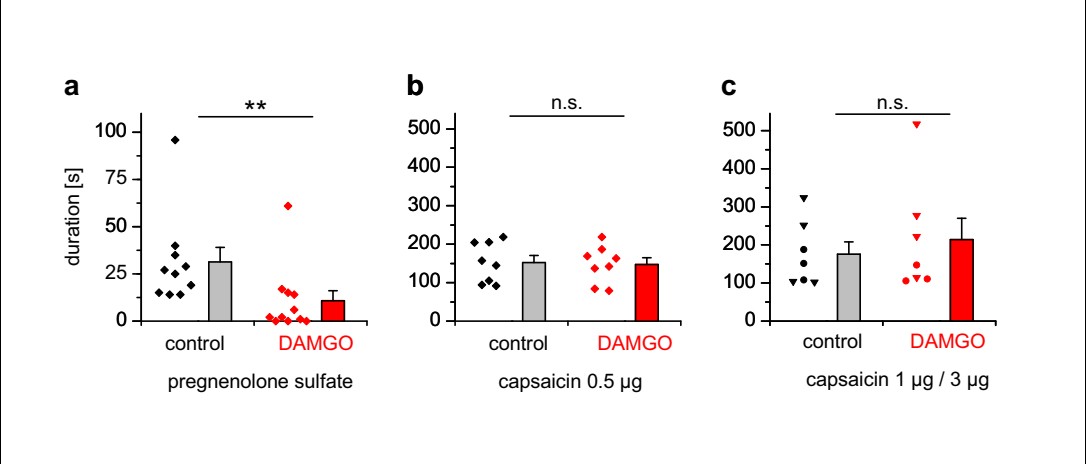

**Figure 10.** PS-induced, but not capsaicin-induced pain is reduced by co-injected μOR agonists. (**a**) Mice were injected 5 nmol PS either together with vehicle as control or with 2 μg DAMGO. The duration of nocifensive behavior was analyzed (n = 10 mice for the control condition and n = 11 mice for the DAMGO condition). (**b**) Instead of the TRPM3 agonist, 0.5 μg capsaicin was used as TRPV1 agonist (n = 8 mice in each column). (**c**) An increased concentration of capsaicin (1 μg, three mice, round symbols, or 3 μg, four mice, inverted triangles) was used. To increase the statistical power, the data of the two capsaicin concentrations were combined.
DOI: https://doi.org/10.7554/eLife.26280.032

nociception (*Nockemann et al., 2013*). Other ion channels seem to be regulated by peripheral μORs via their effects on the cellular cAMP level, often through subsequently influencing PKA activity (*Ingram and Williams, 1994*; *Gold and Levine, 1996*). This implies that μOR activity only affects these targets when cellular cAMP levels are elevated, for instance during inflammatory conditions. Notably, TRPV1 channels are also regulated by μORs through the well-established cAMP/PKA pathway (*Vetter et al., 2006*; *Endres-Becker et al., 2007*; *Vetter et al., 2008*; *Spahn et al., 2013*). Since we were studying TRPV1 channels in DRG neurons under resting, non-inflamed conditions, these considerations might explain why we observed no or only very modest effects of μOR activation on TRPV1 channel activity (*Figure 4a,b,e*; *Figure 4—figure supplement 1*). An entirely different, β-arrestin2-dependent pathway leading from μOR stimulation to TRPV1 activation has also been proposed (*Rowan et al., 2014*). Remarkably, and in stark contrast to the published regulation of TRPV1 by cAMP/PKA (*Huang et al., 2006*), TRPM3 channels were largely unaffected by our attempts to manipulate intracellular cAMP levels (*Figure 5e–h*).

These results from our cellular studies are well matched by the outcome of the in vivo experiments. TRPM3-dependent pain (evoked by the injection of PS into the hind-paw) was strongly suppressed by concomitant injection of μOR agonists, while these substances did not significantly attenuate TRPV1-dependent pain due to the injection of capsaicin at any of the concentrations tested (*Figure 10*). Together, these data demonstrate that the strong and direct functional influence of activated μORs on TRPM3 channels also works in peripheral endings of primary nociceptive neurons in vivo. Our findings establish TRPM3 channels as privileged target of peripheral μORs and thus indicate that TRPM3 channels play an important role in the physiological control of nociceptor excitability.

## TRPM3 is a candidate target for anti-nociceptive pharmacological interventions

The presence of TRPM3 channels in small-diameter nociceptor neurons (*Vriens et al., 2011*; *Usoskin et al., 2015*), their ability to release inflammatory mediators like CGRP (*Held et al., 2015*), some aspects of their pharmacology (*Straub et al., 2013b*; *Chen et al., 2014*; *Suzuki et al., 2016*) and the phenotype of TRPM3-deficient mice, which strongly implied TRPM3 channels in the sensation of noxious heat and in inflammatory heat hyperalgesia (*Vriens et al., 2011*), have previously advanced the argument that inhibition of TRPM3 might be a viable strategy to combat pain,

especially inflammatory hyperalgesia. The results presented in this study provide further, strong support for this contention, because they show that TRPM3 inhibition is an important aspect of the action spectrum of µORs. Potentially, the findings presented here therefore help to explain the clinical effectiveness of peripherally acting or peripherally restricted µOR agonists. However, peripherally restricted µOR agonists still can exhibit pronounced and dose-limiting adverse effects, such as tolerance and constipation (*Stein, 2016*). Pharmacologically inhibiting TRPM3 channels directly might therefore be a feasible alternative to the established administration of µOR agonists. Given the unremarkable phenotype of TRPM3-deficient mice, when not challenged with painful stimuli, (*Vriens et al., 2011*), it is reasonable to hope that unwanted effects of TRPM3 inhibitors may be less limiting than those of µOR agonists. It will be important to elucidate in various painful conditions how strong and robust the relief is that can be obtained by antagonists of TRPM3 channels. This study provides a strong incentive for commencing such work.

## Materials and methods

### Animals

For cellular experiments, we used adult mice of a wide age range (aged 12–77 weeks) of both sexes. The animals were either C57BL/6 mice or TRPM3-deficient mice (*Vriens et al., 2011*) that were back-crossed for more than 10 generations to the C57BL/6 genetic background. The TRPM3-deficient mouse strain is a classical, unconditional knock-out strain, in which exon 19 has been substituted by a LacZ-neomycin cassette. Housing and killing of the animals were carried out with institutional approval and in compliance with the guidelines of the Regierungspräsidium Gießen (AK-3–2014). For behavioral experiments, only male C57BL/6 mice at an age of 7–9 weeks were used. These experiments were approved and carried out in compliance with institutional guidelines of the Max Planck Society and guidelines of the Landesamt für Verbraucherschutz und Lebensmittelsicherheit of Lower Saxony, Germany (AZ 33.9-42502-04-14/1638).

### Isolation and culture of dorsal root ganglion neurons

Mice were killed by an overdose of isoflurane (5%, AbbVie, Wiesbaden, Germany) and then decapitated. The spinal cord was exposed by a single dorsal-midline incision along the entire length of the mouse. The entire spinal cord was removed, washed and placed into ice-cold HBSS (GIBCO, Thermo Fisher, Karlsruhe, Germany). The spine was bisected along the spinal canal in longitudinal direction, nerve trunks and connective tissue was removed. Dorsal root ganglia (DRGs) from all cervical, thoracic and lumbar segments were then harvested into ice-cold culture medium consisting of DMEM (GIBCO) supplemented with 10% fetal calf serum (GIBCO), and 1% penicillin-streptomycin (100 U/ml and 100 µg/ml, GIBCO). Isolated ganglia were partially digested for 30 min in 1.8 U/ml liberase DH Research Grade (Roche, Mannheim, Germany) at 37°C. DRGs were then gently triturated with a 1000 µl pipette. The digestion was stopped by adding 10–12 ml culture medium and centrifugation of the dissociated DRGs for 5 min at 250 g; washing and centrifugation was typically repeated for a second time. The supernatant was discarded and the cells were suspended in culture medium. Subsequently, for $Ca^{2+}$ imaging experiments one eighth of the cell suspension (corresponding to 100 µl) was plated onto the center of a glass coverslip pre-coated with laminin (Sigma-Aldrich, Munich, Germany). The cells were left to adhere at 37°C in an incubator in a humidified atmosphere containing 5% $CO_2$. After 2 hr, 2 ml culture medium was added onto the coverslips. For electrophysiological experiments, 100 µl of the cell suspension was diluted with 2 ml culture medium and seeded as such in a laminin-coated plastic culture dish (Falcon, VWR, Darmstadt, Germany). Cells were maintained in the incubator and all experiments were performed within 24–56 hr after plating the cells.

### Cell culture of cell lines

Human embryonic kidney 293 (HEK) cells, HEK-TRPM3 cells, which stably express either myc-TRPM3α2 (*Frühwald et al., 2012*) or myc-TRPM3α2-YFP (*Oberwinkler et al., 2005*) and HEK cells stably expressing human TRPM8 (*Erler et al., 2006*), kindly provided by Dr. U. Wissenbach (Homburg, Germany), were cultured and handled as described previously (*Wagner et al., 2008*; *Frühwald et al., 2012*; *Drews et al., 2014*). Alternatively, we used HEK cells transiently transfected with TRPM3α2 as described (*Wagner et al., 2008*). Neither in this work, nor in our previous studies,

did we observe differences in the TRPM3 channel properties (apart from transfection efficiencies) due to transfection methods (whether transiently or stably) or terminal protein fusion tags employed (see also *Figure 8—figure supplement 1*). TRPM3 proteins exist in many different isoforms, mainly due to alternative splicing (*Lee et al., 2003*; *Oberwinkler et al., 2005*; *Frühwald et al., 2012*; *Oberwinkler and Philipp, 2014*). Here, we use the naming of the splice variants according to *Oberwinkler and Philipp (2014)*. Throughout this study, we refer to heterologously expressed TRPM3$\alpha$2 and to TRPM3 channels endogenously expressed in DRG neurons (which have not been characterized with respect to splice events) as TRPM3. HEK cells and derivative cell lines were grown in MEM (GIBCO) supplemented with 10% fetal calf serum. Geneticin (0.5 mg/ml, Sigma-Aldrich) was added to the culture medium for stably transfected cells only. CHO-TRPA1 cells, i.e. chinese hamster ovary cells stably expressing mouse TRPA1 (*Story et al., 2003*), kindly provided by Dr. A. Patapoutian (San Diego, USA), were cultured in DMEM (GIBCO) supplemented with 10% fetal calf serum (GIBCO), 1% penicillin-streptomycin (GIBCO), 10 mM glutamax (GIBCO) and 10x non-essential amino acids (GIBCO). As we used these cell lines merely and exclusively as containers for expression of ion channels and other signaling molecules, we did not routinely test the identity of these cell lines. We routinely and regularly tested, however, the expression of the stably transfected genes, both by western blotting and, more often, by functional tests (Ca$^{2+}$ imaging and/or electrophysiology) to ascertain the presence of the stably transfected ion channels (which the parental cell lines do not express). These tests also ensure that the stably transfected cell lines are not mis-identified. All cell lines were maintained at 37°C in a humidified atmosphere with 5% CO$_2$. Cells were passaged one to three times per week, care was taken to avoid passage numbers above 40. For experiments, HEK and CHO cells (and cells from derived lines) were plated on coverslips coated with poly-L-lysine (MW: 70,000; Sigma-Aldrich).

## Transfection and expression vectors

Transient transfection of HEK cells (or derived cells lines, see above) and CHO-TRPA1 was achieved with PolyFect (Qiagen, Hilden, Germany) according to the manufacturer's instructions. Measurements were performed 24–72 hr after transfection. When cells were used for electrophysiological experiments, they were typically passaged 1 day before measurement to reduce their density.

The following constructs and expression vectors were used. Human µOR in pcDNA3.1 was purchased from the cDNA Resource Center (Bloomsburg, PA, USA). Starting from this vector, we generated the fusion construct human µOR-YFP (in pcDNA3) with standard procedures and oligonucleotides obtained from Eurofins (Ebersberg, Germany). Rat TRPV1-YFP in pcDNA3 (*Hellwig et al., 2005*) was obtained from Dr. T. Plant (Marburg, Germany). TRPM3 proteins were expressed with the help of vectors containing wild-type TRPM3$\alpha$2, myc-TRPM3$\alpha$2, YFP-TRPM3$\alpha$2 or TRPM3$\alpha$2-YFP either in pcDNA3 or in pCAGGS, which contained an additional IRES-GFP sequence enabling easy identification of transfected cells. All TRPM3 expression vectors encoded for murine TRPM3 proteins. All wild-type G$\alpha_i$ cDNAs without tag (human G$\alpha_{i1}$, G$\alpha_{i2}$ and G$\alpha_{i3}$) were obtained in pcDNA3.1 vectors from the cDNA Resource Center. Vectors (either pcDNA3 or pCAGGS) containing human G$\alpha_{i1}$Q204L (here named G$\alpha_{i1}$QtoL), G$\alpha_{o1}$, rat G$\alpha_{o1}$Q205L (here named G$\alpha_{o1}$QtoL), G$\alpha_q$QtoL and bovine G$\gamma_2$ (which has the same amino acid sequence as human and mouse G$\gamma_2$) were kindly provided by Dr. M.X. Zhu (Houston, USA). Rat G$\alpha_{i1}$-YFP, G$\alpha_{i2}$-YFP and G$\alpha_{i3}$-YFP in pcDNA3 and human G$\beta_1$ (having the same amino acid sequence as mouse G$\beta_1$) in pCMV were described previously (*Bünemann et al., 2003*; *Frank et al., 2005*). Human G$\alpha_{oB}$ (here named G$\alpha_{o2}$), G$\alpha_{oB}$Q205L (here named G$\alpha_{o2}$QtoL), FLAG-G$\beta_1$ and G$\alpha_q$Q209L (here named G$\alpha_q$QtoL) were from the cDNA Resource Center (all in pcDNA3.1). Bovine $\beta$ARKct (*Koch et al., 1994*) and mouse phosducin (kindly provided by Dr. L. Hein, Freiburg, Germany) were subcloned into the expression vector pCAGGS (with the additional IRES-GFP cassette) and N-terminal myristoylation tags were added (*Rishal et al., 2005*) with standard procedures. Specifically, we added a sequence encoding the first 15 amino acids of Src followed by the triplet GAT (encoding for aspartate) as a linker before the start codon of the original proteins. For control transfections we used empty pcDNA3 or pCAGGS-IRES-GFP vectors. To visually identify successfully transfected cells we co-transfected pcDNA3-IRES-GFP or ER-DsRed (kindly provided by Dr. R. Jacob, Marburg, Germany) when the other transfected plasmids did not express a fluorescent protein. Verification of DNA sequences was done by direct sequencing (Seqlab, Göttingen, Germany).

## Solutions for functional cellular studies

The concentrations indicated throughout this section are the final values after adjustment of pH. The standard extracellular solution contained (in mM): 145–149 NaCl, 10 CsCl, 3 KCl, 2 CaCl$_2$, 2 MgCl$_2$, 10 HEPES, 3 D-glucose, 7 D-mannitol (pH 7.2). In some instances the D-mannitol was replaced by 7 mM D-glucose. The extracellular solution with an elevated concentration of K$^+$ (high-K$^+$ solution) contained (in mM): 70–74 NaCl, 10 CsCl, 75 KCl, 2 CaCl$_2$, 2 MgCl$_2$, 10 HEPES, 10 D-glucose. These extracellular solutions were adjusted with NaOH to pH 7.2. The osmolality was regulated to within 315–335 mOsm/kg by the addition of D-glucose or H$_2$O. The monovalent-free extracellular solution contained (in mM): 2 CaCl$_2$, 2 MgCl$_2$, 10 HEPES, 280 D-mannitol. NMDG was used here to adjust the pH to 7.4, resulting in 4–5 mM NMDG in the solution. Osmolality was 312–316 mOsm/kg.

Standard intracellular solution for patch-clamping contained (in mM): 140–145 CsOH, 10 BAPTA, 50 CsCl, 80 aspartate, 4 Na$_2$ATP, 3 MgCl$_2$, 10 HEPES. The ATP-free intracellular solution contained (in mM): 140–145 CsOH, 10 BAPTA, 50 CsCl, 80 aspartate, 4 Li$_4$AMP-PNP, 1 Na$_2$GTP, 1 MgCl$_2$, 10 HEPES. The Mg$^{2+}$-free intracellular solution contained in (mM): 135 CsOH, 80 aspartate, 50 CsCl, 10 BAPTA, 10 HEPES, 5 Na$_2$EDTA, 4 Na$_2$ATP. The pH of all intracellular solutions was adjusted to 7.2 with CsOH and the osmolality of these solutions was in the range of 285–315 mOsm/kg.

For the recordings shown in *Figure 2d–f*, we used solutions without Cs$^+$ ions, in order to avoid inhibition of K$^+$ channels. The Cs$^+$-free extracellular solution contained (in mM): 144 NaCl, 5.8 KCl, 0.9 MgCl$_2$, 1.3 CaCl$_2$, 0.7 NaH$_2$PO$_4$, 5.6 D-glucose and 10 HEPES, with a pH of 7.4 (adjusted with NaOH) and an osmolality of 305 mOsm/kg. The Cs$^+$-free intracellular solution was composed of (in mM): 135 KCl, 3.5 MgCl$_2$, 2.4 CaCl$_2$, 5 EGTA (resulting in a free Ca$^{2+}$ concentration of 100 nM), 5 HEPES, 2.5 Na$_2$ATP. The pH of this solution was adjusted to 7.3 with KOH, the osmolality was 285 mOsm/kg.

## Calcium imaging

Intracellular Ca$^{2+}$ imaging was performed as described previously (*Drews et al., 2014*). Generally, coverslips with the cells attached were incubated with 5 µM Fura2-AM (1 mM stock in DMSO, Biotrend, Cologne, Germany) for 30 min in the respective culture medium. Loading and measurements took place at room temperature (22–25°C), except where indicated otherwise (see *Figure 1j*). After loading, coverslips were transferred to a closed recording chamber (Warner Instruments, Hamden, CT, USA) and continuously perfused with standard extracellular solution. Alternatively (for the experiments shown in *Figure 7—figure supplement 3*), we imaged cells in an open chamber containing a small volume (300 µl) of static (not perfused) solution containing the Gβγ-liberating peptide mSIRK or its inactive control mSIRK-L9A (*Goubaeva et al., 2003*). At the indicated time point in the figure, an additional volume of 300 µL of the peptide containing extracellular solution, plus double-concentrated TRPM3-agonist was added manually to the pre-existing volume to ensure fast and efficient mixing. At the end of the experiments, further 600 µl of high-K$^+$ solution was added (raising the average K$^+$ concentration to 39 mM) in order to depolarize the cells and to identify neurons.

During imaging, every 5 s a pair of images was taken at 510 nm wavelength with a Retiga-Exi (Qimaging, Surrey, BC, Canada) or a HQ2 camera (Photometrics, Tucson, AZ, USA) during alternating excitation at 340 and 380 nm wavelengths (filters and dichroic mirrors from AHF, Tübingen, Germany) using a motorized filter wheel (Ludl, Hawthorne, NY, USA) or a wavelength switcher (DG4, Sutter, Novato, CA, USA) attached to Nikon (Düsseldorf, Germany) inverted microscopes equipped with 10x SFluor objectives (N.A. 0.5). From a fluorescence image with 380 nm excitation or a fluorescence image of the GFP/YFP fluorescence, several regions of interest representing each a single cell were selected manually (see *Figure 1—figure supplement 5*). Ratio images (340/380 nm) were calculated with ImageJ (*Abràmoff et al., 2004*) using a modified version of the 'ratio plus' plug-in after background subtraction and thresholding to exclude pixels with low fluorescence intensity values.

After imaging for approximately 5 min to establish baseline conditions, ligands were superfused onto the cells as indicated in figures using a gravity-driven perfusion system. For some experiments, cells were pretreated with pharmacological substances (H89, KT5720, BIM, mSIRK or mSIRK-L9A, *Figure 6* and *Figure 7—figure supplement 3*), for 30 min by adding them to the culture medium during Fura2-AM loading. Pertussis toxin (PTX, 100 ng / ml, List Biological Laboratories, Campbell, CA, USA) was added to the culture medium in the incubator for 16–24 hr and washed off the extracellular medium before loading with Fura2-AM. For imaging DRG neurons, 20 µM verapamil was

added to all extracellular solutions to block endogenous voltage-gated $Ca^{2+}$ channels. However, to distinguish neuronal from non-neuronal cells in these cultures, the high-$K^+$ solution was routinely used without verapamil to depolarize the cells at the end of the experiment.

## Electrophysiology

Whole-cell patch-clamping was performed with EPC10 amplifiers (HEKA, Lambrecht/Pfalz, Germany) as described previously (*Drews et al., 2014*). Series resistances were compensated for 80% and all offset potentials were nullified before establishing the cell-attached configuration. All potential values are reported after being corrected for the calculated liquid junction potential (15 mV), except when using the $Cs^+$-free solutions, where we assumed a negligible liquid junction potential. Voltage ramps (−115 to +85 mV with a steepness of 1 mV/ms) were applied with a frequency of 1 ramp per 1–2 s. For HEK and transfected HEK cells a holding potential of −15 mV was used between the ramps and the current amplitudes were analyzed at −80 and +80 mV offline. DRG neurons were voltage-clamped at holding potentials of −55 to −75 mV between the ramps and only the currents at −80 mV were analyzed. In an attempt to diminish endogenous currents during voltage-ramps, the standard extracellular recording solution was replaced by a monovalent-free extracellular solution (*Vriens et al., 2011*; *Straub et al., 2013a*) after establishing the whole-cell configuration (*Figure 2a–c*). Alternatively, voltage ramps from −100 to −20 mV were recorded from DRG neurons in $Cs^+$-free intra- and extracellular solution, and subsequently analyzed at −60 mV (*Figure 2d–f*). From DRG neuron recordings used for *Figure 2a–c*, we only considered measurements, in which the application of TRPM3 agonists (50 μM PS + 50 μM nifedipine) resulted in inward currents with an amplitude of more than 10 pA, indicating robust expression of TRPM3 channels. For *Figure 2d–f*, we used recordings from DRG neurons that showed currents > 10 pA to the application of the same TRPM3 agonists or to capsaicin, in an attempt to record from a broad population of small diameter neurons (*Vriens et al., 2011*; *Tan and McNaughton, 2016*). Current densities were calculated offline with the help of the Igor software package (version 5.05A, Wavemetrics, Lake Oswego, OR, USA).

## Chemical reagents

All chemical reagents were prepared as stock solutions in DMSO, with the exception of PTX (pertussis toxin from *Bordetella pertussis*) and morphine that were dissolved in $H_2O$. Stocks were kept aliquoted and frozen at -20°C. Even when several compounds were applied simultaneously, the final DMSO concentration in the superfusing solution did not exceed 0.4%. The following substances were obtained from Sigma-Aldrich: AITC (allyl-isothiocyanate), capsaicin, DAMGO ([D-Ala2, N-Me-Phe4, Gly5-ol]-enkephalin acetate), menthol, morphine sulphate pentahydrate, IBMX (3-Isobutyl-1-methylxanthine), naloxone hydrochloride dihydrate, nifedipine, verapamil hydrochloride. The following substances were purchased from Biotrend: forskolin, herkinorin, loperamide hydrochloride, (RS)-baclofen, WIN 55,212-2, somatostatin-14, DHPG (RS-3,5 dihydroxyphenylglycine), [D-Ala2] deltorphin II. L-noradrenaline was obtained from Alfa Aesar (Thermo Fisher), BIM (Bisindolylmaleimide IV) was from Biomol (Hamburg, Germany), H89 dihydrochloride hydrate from Biozol (Eching, Germany), KT5720 from Santa Cruz (Heidelberg, Germany) and pregnenolone sulfate (PS) from Steraloids (Newport, RI, USA). The cell-permeable peptides mSIRK and mSIRK-9LA were purchased from Calbiochem (Merck-Millipore, Darmstadt, Germany).

## Data analysis, representation and statistical testing

$Ca^{2+}$ imaging time series of single cells were extracted from stacks of ratio images as averages over the entire cell area. The first 20 data points (corresponding to the first 100 s of the experiment) were averaged to form the baseline which, for quantitative analyses, was subtracted from all other values. For quantitative analysis and statistical testing, single cell time series traces were smoothed (running window of five values) and maximum, average or minimum values were obtained in a given time-window (typically corresponding to the application of a substance) from the smoothed single cell $Ca^{2+}$ imaging traces. Inhibition due to a pharmacological manipulation was calculated for $Ca^{2+}$ imaging (and electrophysiological) data by obtaining three (baseline-subtracted) values: One before ($V_{before}$), one during ($V_{during}$) and one after the application ($V_{after}$) of the pharmacological substance. Inhibition (in percent) was calculated as: Inhibition = $100 * (1 - (2 * V_{during} / (V_{before} + V_{after}))$. The averaging of the values before and after pharmacological interventions was done in an effort to correct for the

pronounced and inevitable reduction in response during repeated or prolonged applications of agonists, such as PS or capsaicin. Please note, however, that this way of calculating inhibition (which is a very common and standard way) still is prone to artefacts and can report inaccurate values. For example, a cell that responded to the first application of an agonists ($V_{before}$), but then simply ceased to respond to any further stimulation ($V_{during}$ and $V_{after}$ would then be '0'), would be reported as 100% inhibited. Please note also that the formula for calculating inhibition can report negative values, when a cell responds stronger during the application of the pharmacological intervention ($V_{during}$ larger than the average of $V_{before}$ and $V_{after}$). Because DRG neurons are a highly diverse population and sometimes are also prone to spontaneous activity, the individual values reported in the 'dot clouds' often contain a minority of cells that displayed such negative inhibitory values. Another common source for artificial values is baseline shift, which we did not attempt to correct for.

Isolated DRG cells were classified as neurons if they responded either to capsaicin or the solution with a high potassium concentration (high-$K^+$ solution), cells outside of this category were not considered further. For sorting individual DRG neurons into different categories (e.g. PS-sensitive or capsaicin-sensitive), a threshold of 0.1 ratio units was set arbitrarily (see *Figure 1—figure supplement 1* for a justification of this value). As an exception, in *Figure 7—figure supplement 3a,b*, we have taken a threshold value of 0.05, because the treatment with mSIRK has lowered the response amplitude to PS. For categorizing sensitivity to an inhibitory stimulus, a minimal inhibition of 7.5% of the agonist-induced response (calculated as above) was arbitrarily set as threshold. Traces of single cell time series corresponding to a category were averaged and are presented as mean ± SEM throughout this manuscript. Next to the representation of aggregated data, we also plotted the values from all individual neurons analyzed as 'dot cloud'. For each reported average trace, we also show responses from ten randomly chosen neurons in the figure supplements. Calculating the response values, categorizing, averaging and plotting the randomly chosen cells was done in R.

For $Ca^{2+}$ imaging experiments of (transfected) cell culture cells, we did not categorize individual cells as responders or non-responders, but took all cells according to transfection status or pharmacological treatment. Since these cells represent a much more homogenous group, we did not report single cell values, but only averaged responses.

Cell diameters were estimated by drawing elliptical selections manually around single DRG neurons on ratio images during a strong stimulation (in this case with high-$K^+$ solution). Subsequently, Feret's diameter was measured with the built-in function of ImageJ. The distribution of single cell Feret's diameters was represented as histograms with a bin width of 2 μm.

For $Ca^{2+}$ imaging and electrophysiological experiments, we considered single individual cells as independent biological replicates. We additionally report the number of recordings for $Ca^{2+}$ imaging experiments, which indicates the number of coverslips (with the attached cells) that were used in these experiments in sequential measurements. In electrophysiological experiments, we equally used the number of individual cells that were successfully recorded as the number of biological replicates. Each single, individual co-immunoprecipitation experiment was visualized on a single western blot, which was quantitatively analyzed and considered a biological replicate. Finally, in behavioral experiments, each individual mouse was recorded only once and considered a biological replicate. In all types of experiments, the number of biological replicates (determined as described above and referred to as 'n') was used for statistical testing and to calculate SEM values.

We did not perform any a-priori estimation of sample sizes. We followed, however, closely the standards in the field, and, accordingly, used 7–15 individual biological replicates for each data point in electrophysiological (whole-cell patch-clamping), Western blotting and behavioral experiments. Such numbers for replicates mean that we were able to detect (with a power of 80% and at a level of significance of $p<0.05$) differences between means that were in the order of 1–2 SD. For $Ca^{2+}$ imaging experiments, we always performed at least two separate recordings. During each recording, we typically were able to record at least from 10 cells (depending on parameters such as transfection efficiency and cell survival during pharmacological treatments), meaning that each trace in $Ca^{2+}$ imaging recordings represents the mean of more than 20 cells. The precise number of cells is stated in the figures themselves or, alternatively, in the figure legends.

Because of the small sample sizes in some experiments, and because we observed that many of the larger data sets did not conform to Gaussian normality (as tested with Graphpad Prism, version 3.02, Graphpad software, La Jolla, CA, USA), typically due to outliers (which were never removed from any data set), we preferred non-parametric statistical tests. For comparing two groups, we

used the Mann-Whitney test (*Figure 1c*, *Figure 1—figure supplement 2c*, *Figure 4b*, *Figure 5b,d*, *Figure 7g,i*, *Figure 8c*, *Figure 10a–c*) or the Wilcoxon signed rank test (*Figure 2b,e*). For multiple comparisons, we used either the Kruskal-Wallis test (*Figure 5k*, *Figure 5—figure supplement 1b*, *Figure 7—figure supplement 2*, *Figure 7—figure supplement 3a,c*) or the Friedman test (*Figure 1m*). These two latter tests were followed by Dunn's multiple comparison test. All statistical testings were done with Graphpad Prism or R. In all cases, we accepted p-values smaller than 0.05 as statistically significant. In the figures, we use * to indicate p-values larger or equal than 0.01 and smaller than 0.05, ** for p-values larger or equal than 0.001 and smaller than 0.01 and, accordingly, *** for p-values smaller than 0.001. Unless otherwise stated in the figure legend, bar graphs with error bars always represent the mean ± SEM. Fitting Hill functions (with variable slope) to dose-response curves was done with Graphpad Prism.

## Protein immunoprecipitation and western blots

For immunoprecipitation of TRPM3, HEK cells stably expressing myc-TRPM3α2-YFP (*Oberwinkler et al., 2005*) were solubilized in a lysis buffer consisting of 25 mM Tris (pH 8), 0.5% (w/v) Triton X-100, 0.5% (w/v) Na⁺-deoxycholate, 50 mM NaCl and a protease inhibitor mix as described (*Leitner et al., 2016*). All reagents for immunoprecipitation and western blotting were obtained from Carl Roth (Karlsruhe, Germany) unless stated otherwise. Non-transfected HEK cells were used as controls. After removal of debris and nuclei by centrifugation for 15 min at 13,000 g and 4°C, a known fraction of the lysate (typically 1% of the total lysate) was separated for later use as input control and the remaining lysate was incubated with GFP-Trap agarose beads (ChromoTek, Planegg-Martinsried, Germany) at 4°C over night on a tube roller. Subsequently, beads were washed once with lysis buffer and four times with binding buffer (*Hu et al., 2009*) containing 20 mM HEPES (pH 7.4), 0.01% (w/v) CHAPS, 140 mM K⁺-aspartate, 5 mM MgCl₂, 10 mM EGTA and 0.04 mM dithiothreitol. Input and immunoprecipitated proteins were separated by SDS-PAGE and transferred to nitrocellulose membranes (GE Healthcare, Solingen, Germany). For protein visualization, we used fluorescence (ODYSSEY Sa, LI-COR Biosciences, Bad Homburg, Germany) or chemiluminescence (ChemoCam Imager, Intas, Göttingen, Germany) detection systems together with the following antibodies: anti-GFP (Santa Cruz, sc-8334, 1:500), anti-Gβ (Santa Cruz, sc-378, 1:500), anti-Gα_{i3} (1:1000; [*Gohla et al., 2007*]), anti-Gα_{q/11/14} (Santa Cruz, sc-365906, 1:500), anti-rabbit IgG-IRDye-800CW (LI-COR, #926–32211, 1:10,000), anti-mouse IgG-HRP (Santa Cruz, sc-2031, 1:10,000), anti-rabbit IgG-HRP (Santa Cruz, sc-2030, 1:10,000). Immunoprecipitation experiments were quantified densitometrically using ImageJ (*Abràmoff et al., 2004*), the density values for the immunoprecipitated Gβ proteins were subsequently normalized to the density values obtained for Gβ proteins from the total lysate.

## Mass spectrometry

TRPM3 protein isolation was performed essentially as described previously (*Uchida et al., 2016*). In brief, seven to eight 10 cm culture dishes with HEK cells stably expressing myc-TRPM3α2 (*Uchida et al., 2016*) were grown to ~80% confluence and used for a single immunoprecipitation probe. Cells were washed and collected with cold phosphate buffered saline (PBS) and resuspended in NCB buffer, containing (in mM): 500 NaCl, 50 NaH₂PO₄, 20 HEPES and 10% (v/v) glycerol (pH 7.5), with addition of 1 mM protease inhibitor phenyl-methyl-sulfonyl-fluorid and 5 mM β-mercaptoethanol. Next, the cells were lysed by freezing/thawing and centrifuged at 40,000 g for 2.5 hr. The pellet was resuspended in NCB buffer with the addition of a protease inhibitor mixture (Roche), 0.1% (w/v) Nonidet P-40 (Roche) and 0.5% (w/v) dodecylmaltoside (Calbiochem). The suspension was incubated overnight on a shaker with gentle agitation, and then centrifuged for 1 hr at 40,000 g. The supernatant was incubated with magnetic beads (Pierce, Thermo Fisher) conjugated with anti-myc antibodies (Sigma-Aldrich). All steps of incubation were performed at 4°C. TRPM3 proteins were eluted from the beads with SDS-loading buffer by boiling. The eluted proteins then were separated by SDS-PAGE on 10% polyacrylamide gels and Tris-glycine-SDS buffer (Bio-Rad, Hercules, CA, USA) at a constant voltage of 180 V. Proteins were visualized by Coomassie blue staining.

Besides the heavy and light chain antibody bands derived from immunoprecipitation, the gel showed a profound band at approx. 212 kDa, corresponding to the molecular weight of TRPM3 monomers. From each gel (corresponding to one experiment), 12–14 bands were excised and

digested with trypsin for mass spectrometry analysis according to a published protocol (*Shevchenko et al., 2006*), with some modifications. The protein digests were analyzed by liquid chromatography-tandem mass spectrometry (LC-MS/MS) using a nano flow liquid chromatography system (Ultimate3000, Thermo Fisher) interfaced to a hybrid ion trap-orbitrap high resolution tandem mass spectrometer (VelosPro, Thermo Fisher) operated in data-dependent acquisition mode, as previously described (*Uchida et al., 2016*). Data analysis were performed on ProteomeDiscoverer 2.1 (Thermo Fisher) using SequestHT (0.1% false discovery rate) (*Tabb, 2015*) and Percolator for peptide/protein identification and validation (*Käll et al., 2007*).

### Behavioral testing

After acclimatization for at least 1 hr in transparent plastic boxes, the mice were injected subcutaneously with 10 µl of the respective experimental agent into the midplantar region of the left hind paw. The total duration of nocifensive behavior (paw lifting, shaking and licking) during a 20-min observation period was measured. For PS-induced pain mice received either 5 nmol PS together with 2 µg DAMGO, both dissolved in PBS or 5 nmol PS and the vehicle. For dissolving PS in PBS, the solution was heated to 30°C and sonicated for 20 min. For capsaicin-induced pain, mice were injected with 0.5 µg capsaicin (dissolved in 3% ethanol/PBS) together with 2 µg DAMGO or 0.5 µg capsaicin and vehicle (PBS). Experiments were done with a minimum of three independent mouse litters. Whenever possible, littermates were used as controls (vehicle injection) on the same experimental day.

## Acknowledgements

The authors thank D Wagner, S Plant and M Portz for excellent technical support during different phases of this project. We also thank Drs L Hein (Freiburg, Germany), R Jacob (Marburg, Germany), A Patapoutian (San Diego, USA), T Plant (Marburg, Germany), U Wissenbach (Homburg, Germany) and MX Zhu (Houston, USA) for the generous gift of plasmids and cell lines. We also thank Drs L Hein and MX Zhu for advice at early stages of this study.

## Additional information

### Funding

| Funder | Grant reference number | Author |
| --- | --- | --- |
| Deutsche Forschungsgemeinschaft | Emmy Noether Programm | Manuela Schmidt |
| Deutsche Forschungsgemeinschaft | GK 1326 | Johannes Oberwinkler |
| Max-Planck-Gesellschaft | PhD fellowship | Julia Sondermann |
| Philipps-Universität Marburg | MARA PhD fellowship | Christian Goecke |
| Deutsche Forschungsgemeinschaft | SFB 593 | Johannes Oberwinkler |
| Deutsche Forschungsgemeinschaft | SFB 894 | Johannes Oberwinkler |

The funders had no role in study design, data collection and interpretation, or the decision to submit the work for publication.

### Author contributions

Sandeep Dembla, Conducted imaging experiments, Analyzed data, Wrote, reviewed and edited the manuscript; Marc Behrendt, Conducted imaging experiments, Conducted electrophysiological experiments, Conducted Western blotting experiments, Analyzed data, Wrote, reviewed and edited the manuscript; Florian Mohr, Conducted imaging experiments, Conducted electrophysiological experiments, Analyzed data, Reviewed and edited the manuscript; Christian Goecke, Franziska M Schneider, Marlene Schmidt, Conducted electrophysiological experiments, Analyzed data, Reviewed

and edited the manuscript; Julia Sondermann, Conducted mouse behavioral experiments, Reviewed and edited the manuscript; Julia Stab, Raissa Enzeroth, Paulina Nuñez-Badinez, Wolfgang Greffrath, Conducted imaging experiments, Reviewed and edited the manuscript; Michael G Leitner, Dominik Oliver, Conducted electrophysiological experiments, Reviewed and edited the manuscript; Jochen Schwenk, Bernd Nürnberg, Stephan E Philipp, Moritz Bünemann, Contributed reagents and material, Reviewed and edited the manuscript; Alejandro Cohen, Analyzed data, Reviewed and edited the manuscript; Eleonora Zakharian, Conducted experiments on protein isolation for mass spectrometry, Analyzed data, Reviewed and edited the manuscript; Manuela Schmidt, Designed and supervised mouse behavioral experiments, Analyzed data, Reviewed and edited the manuscript; Johannes Oberwinkler, Designed the study, Analyzed data, Supervised imaging and electrophysiological experiments, Wrote, reviewed and edited the manuscript

### Author ORCIDs

Manuela Schmidt http://orcid.org/0000-0003-1972-3519
Johannes Oberwinkler http://orcid.org/0000-0002-8541-3626

### Ethics

Animal experimentation: All animal experimentation was approved and carried out in strict compliance with institutional guidelines of the Max Planck Society and guidelines of the Landesamt für Verbraucherschutz und Lebensmittelsicherheit of Lower Saxony, Germany (AZ 33.9-42502-04- 14/1638)

### Decision letter and Author response

Decision letter https://doi.org/10.7554/eLife.26280.034
Author response https://doi.org/10.7554/eLife.26280.035

## Additional files

### Supplementary files

• Transparent reporting form
DOI: https://doi.org/10.7554/eLife.26280.033

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
