## [Decision Letter]

Thank you for submitting your article "Anti-nociceptive action of peripheral µ-opioid receptors by Gβγ-mediated inhibition of TRPM3 channels" for consideration by *eLife*. Your article has been reviewed by three peer reviewers, and the evaluation has been overseen by Kenton Swartz as the Reviewing Editor and Richard Aldrich as the Senior Editor. The following individuals involved in review of your submission have agreed to reveal their identity: Thomas Voets (Reviewer #1); László Csanády (Reviewer #2); Alexander Chesler (Reviewer #3).

The reviewers have discussed the reviews with one another and the Reviewing Editor has drafted this decision to help you prepare a revised submission.

Summary:

This manuscript is one of three that reports exciting new findings on the mechanism of inhibition of TRPM3 channel activity in dorsal root ganglion (DRG) sensory neurons by stimulation of G protein-coupled receptors (GPCRs). Although all three manuscripts received favorable reviews, the essential revisions will require different amounts of time to address, and we encourage the authors to coordinate submission of their revised manuscripts.

The main conclusion of this study is that inhibition of TRPM3 activity by stimulation of GPCRs (in particular mu-opioid receptors) is mediated through a direct interaction of the channel protein with Gβγ dimers released from G-α subunits of the G_i/o_ subfamily. This finding is a breakthrough in understanding the regulation of TRPM3 channel activity, as well as the involvement of TRPM3 channels in peripheral nociception. Overall this is a very carefully conducted study, which combines Ca^2+^ imaging and whole-cell patch-clamp recordings performed in dissociated DRG neurons, and in a heterologous expression system (HEK-293 cells) in which individual components of various GPCR signaling pathways are overexpressed together with TRPM3. In addition, co-immunoprecipitation is used to demonstrate (or rule out) direct interaction between TRPM3 and various G-protein subunits. Finally, in vivo experiments demonstrate significant reduction in TRPM3-dependent pain by stimulation of peripheral opioid mu receptors. Most figures are accompanied by one or more supplementary figures which illustrate careful control experiments. Overall, this is an exciting and thorough study that is very appropriate for publication in *eLife*.

Essential revisions:

1) Much of the data in paper is from calcium imaging experiments. To fully evaluate the quality and robustness of calcium imaging data it is helpful to have representative images and representative traces. This would allow the reader to appreciate the density and health of the cells and neurons as well as the robustness of individual responses. Additionally, it is unclear to us how the data are averaged/presented and the error bars were confusing. Please consider the presentation and make it clear to the reader what is being presented.

2) The involvement of TRPV1 in mu-opioid mediated inflammatory analgesia (e.g. Maione et al. 2009) is well-established, as mentioned in the Discussion, and it is not clear to what extent this form of analgesia is mediated through TRPM3. We request that the authors reframe the manuscript with a more balanced Introduction and Discussion that includes the other well-known MOR-modulated pathways and places TRPM3 within this broader context.

---

## [Author Response]

*Essential revisions:*
*1) Much of the data in paper is from calcium imaging experiments. To fully evaluate the quality and robustness of calcium imaging data it is helpful to have representative images and representative traces. This would allow the reader to appreciate the density and health of the cells and neurons as well as the robustness of individual responses.*

This comment has given us the incentive to try to present data for each individual DRG neuron that we have analyzed. In order to do so, we had to re-analyze all of the data sets. We now provide:

– A “dot cloud” next to each bar of a bar graph. The bars still represent mean +/- SEM values. In the “dot-clouds”, each point represents a single, individual cell. Therefore these graphs represent faithfully the whole data set and show the variability of the measurements.

– Additionally, we show for each averaged trace ten randomly chosen cells in the same time-series representation (y-axis: fura2 signal, x-axis: time). We have taken ten cells as a compromise to show enough traces to allow an impression of the variability, but not too many traces to still allow following individual traces. Please note that since these cells were randomly chosen, they might not be representative of the mean value after averaging the traces of all cells. However, by choosing the cells randomly, we believe that we have avoided selection biases. These graphs are all in supplement figures.

– For the data set of Figure 2, we also provide so-called “heat maps”, for which the cell responses are ordered and represented in a false color code. This allows visualizing all cells analyzed (614 in this case) without overlap.

– In Figure 1—figure supplement 5, we show a set of representative images of the isolated DRG cells on coverslips, taken under various stimuli and illumination conditions.

Please note that compared to the data submitted previously, we have changed the fura-2 threshold value that we used for categorizing the cells as responders or non-responders (was previously 0.05, we now use 0.1). This change appeared useful, because the single cell analysis allowed us to look at the response statistics of wild-type and TRPM3-deficient (“TRPM3 knock-out”) animals. This analysis is presented in Figure 1—figure supplement 1 and shows that a threshold of 0.05 is too liberal and allows too many false-positive cells in the responder categories. Please note that all our conclusions and statistical tests were confirmed when using the new analysis with the new threshold values (i.e. our analysis turned out to be rather robust), however, obviously, the number of neurons in the “responder” categories is now reduced.

Because responses of (transfected) cell culture lines are much more homogenous, and no categorization is necessary in these experiments, we did not perform the single cell analysis for experiments with cells from cell culture lines.

*Additionally, it is unclear to us how the data are averaged/presented and the error bars were confusing. Please consider the presentation and make it clear to the reader what is being presented.*

We have rewritten the corresponding parts in the Materials and methods section (see paragraph entitled: “Data analysis, representation and statistical testing”). We hope that the new text is better and easier to understand. We have also changed the representation of the error bars on the averaged traces, in as much as we have removed the horizontal “handles” of the error bars. Additionally, we have made the error bars “lighter” in color. This allows now to differentiate visually between the averaged mean value and the SEM values. We hope that these changes have improved our presentation sufficiently.

*2) The involvement of TRPV1 in mu-opioid mediated inflammatory analgesia (e.g. Maione et al. 2009) is well-established, as mentioned in the Discussion, and it is not clear to what extent this form of analgesia is mediated through TRPM3. We request that the authors reframe the manuscript with a more balanced Introduction and Discussion that includes the other well-known MOR-modulated pathways and places TRPM3 within this broader context.*

In the Introduction, we have added a whole paragraph discussing a large part of the known µOR-modulated pathways. By citing recent reviews on this topic, we hope to guide interested readers into this somewhat complex and controversial field, while hoping to maintain readability and conciseness of the Introduction.

We realize that our claim that TRPM3 is a key target of µOR signaling is somewhat surprising. Because we feel that this point is important, we have additionally performed new experiments in which we again tested the effects of DAMGO on TRPV1-dependent responses. These new data fully confirm and extend our previous data, the new data is now shown as Figure 4, the old data are still shown, now as Figure 4—figure supplement 1.